# Nociceptive interneurons control modular motor pathways to promote escape behavior in *Drosophila*

**Anita Burgos[1]\*, Ken Honjo[2], Tomoko Ohyama[3], Cheng Sam Qian[1], Grace Ji-eun Shin[4], Daryl M Gohl[5], Marion Silies[6], W Daniel Tracey[7,8], Marta Zlatic[9], Albert Cardona[9], Wesley B Grueber[1,4,10]\***

[1]Department of Neuroscience, Columbia University Medical Center, New York, United States; [2]Faculty of Life and Environmental Sciences, University of Tsukuba, Tsukuba, Japan; [3]Department of Biology, McGill University, Montreal, Canada; [4]Department of Physiology and Cellular Biophysics, Columbia University Medical Center, New York, United States; [5]University of Minnesota Genomics Center, Minneapolis, United States; [6]European Neuroscience Institute Göttingen, Göttingen, Germany; [7]The Linda and Jack Gill Center for Biomolecular Science, Indiana University, Bloomington, United States; [8]Department of Biology, Indiana University, Bloomington, United States; [9]Janelia Research Campus, Howard Hughes Medical Institute, Ashburn, United States; [10]Mortimer B. Zuckerman Mind Brain Behavior Institute, Columbia University, New York, United States

**\*For correspondence:**
ab3271@columbia.edu (AB);
wg2135@columbia.edu (WBG)

**Competing interests:** The authors declare that no competing interests exist.

**Abstract** Rapid and efficient escape behaviors in response to noxious sensory stimuli are essential for protection and survival. Yet, how noxious stimuli are transformed to coordinated escape behaviors remains poorly understood. In *Drosophila* larvae, noxious stimuli trigger sequential body bending and corkscrew-like rolling behavior. We identified a population of interneurons in the nerve cord of *Drosophila*, termed Down-and-Back (DnB) neurons, that are activated by noxious heat, promote nociceptive behavior, and are required for robust escape responses to noxious stimuli. Electron microscopic circuit reconstruction shows that DnBs are targets of nociceptive and mechanosensory neurons, are directly presynaptic to pre-motor circuits, and link indirectly to Goro rolling command-like neurons. DnB activation promotes activity in Goro neurons, and coincident inactivation of Goro neurons prevents the rolling sequence but leaves intact body bending motor responses. Thus, activity from nociceptors to DnB interneurons coordinates modular elements of nociceptive escape behavior.
DOI: https://doi.org/10.7554/eLife.26016.001

## Introduction

Nociception promotes the avoidance of harmful stimuli, and is a fundamental and evolutionarily conserved somatic sense. Although the sensory neurons that detect noxious stimuli have been well studied in numerous organisms, how noxious stimuli are transformed to the complex sequence of behaviors that protect animals from harm remains poorly understood. A key goal is to integrate anatomical, connectivity, and behavioral data to provide a comprehensive view of nociceptive circuit function.

*Drosophila* larvae provide an advantageous system in which to dissect nociceptive circuit organization, connectivity and function. Dendritic arborization (da) sensory neurons extend axon terminals to discrete locations of the ventral nerve cord in a modality specific manner (*Grueber et al., 2007*;

*Merritt and Whitington, 1995*). The stereotypical projections of da sensory axons, characterization of da neuron function, and accessibility of central neurons afforded by large collections of Gal4 lines (*Gohl et al., 2011*; *Jenett et al., 2012*; *Li et al., 2014*) permit dissection of somatosensory circuit organization.

Class IV (cIV) da neurons are polymodal nociceptive neurons with receptive territories that together tile the entire larval epidermis (*Grueber et al., 2002*; *Hwang et al., 2007*; *Xiang et al., 2010*). cIV neural activity is both necessary and sufficient for generating defensive withdrawal (nocifensive) behavior in response to noxious stimuli (*Hwang et al., 2007*). Strong mechanical and high thermal stimulation induce C-shaped body bending and corkscrew-like lateral turning (rolling) behavior, followed by rapid forward locomotion, or escape crawl (*Hwang et al., 2007*; *Ohyama et al., 2013*; *Tracey et al., 2003*). The behavioral responses of *Drosophila* larvae to noxious stimuli are thus both diverse and sequential, suggesting complexity in the circuits downstream of primary sensory neurons. Electron microscopic (EM) reconstruction of ventral nerve cord circuitry has identified circuit elements downstream of cIV neurons, including Basin and Goro neurons, that integrate vibration and noxious stimuli (*Ohyama et al., 2015*). Basin cells receive multiple sensory inputs in distinct regions of the arbor, and impinge on the command-like rolling Goro interneurons (*Jovanic et al., 2016*; *Ohyama et al., 2015*). Although previous data suggest complexity in transduction and integration of inputs leading to nociceptive behavior, how microcircuits promote and coordinate the rapid induction of sequential stages of nociceptive behavior remains unknown.

Here, we identify a population of somatosensory interneurons, the Down-and-Back (DnB) neurons, which arborize in the nociceptive neuropil, and receive input from both nociceptive and gentle-touch sensory neurons. DnBs are activated by heat stimuli in the noxious range, and their activity promotes C-shaped bending and rolling, but not behaviors associated with gentle touch. EM reconstruction indicates that DnBs receive almost exclusive sensory inputs, and provide major input to premotor neurons and indirect input to Goro rolling command-like neurons. We find that DnBs promote the activity of Goro interneurons, and that DnB-induced rolling, but not C-bending, is dependent on Goro activity. Thus, studies of DnB neurons reveal a sequential and modular organization of escape behavior, and a node in the nociceptive circuit that coordinates essential components of nocifensive behavior to enable rapid escape locomotion.

## Results

### Identification of interneurons that promote nociceptive behavior

To gain access to somatosensory circuitry, we examined integrase swappable in vivo targeting element (InSITE) Gal4 lines (*Gohl et al., 2011*) for expression in the ventral region of the ventral nerve cord (VNC) where class IV (cIV) nociceptive axons terminate (*Grueber et al., 2007*)(*Figure 1A*). We identified two promising lines, *412-Gal4* and *4051-Gal4*, that labeled segmental interneurons with processes in the ventromedial neuropil (*Figure 1B*, *Figure 1—figure supplement 1A*). We generated *412-QF* by standard InSITE swapping methods (*Gohl et al., 2011*) and verified overlapping expression of *4051-Gal4* and *412-QF* in the same population of interneurons in the VNC (*Figure 1—figure supplement 1B–C*). *412-Gal4* also labeled a bilateral population of neurons in the brain lobes, and faintly labeled other cell bodies in the VNC (typically 2, and up to 6, additional cells per hemisegment; *Figure 1B*), but did not label primary sensory neurons or motor axons (*Figure 1—figure supplement 2A–C'*). We identified the two additional neuron types consistently labeled by *412-Gal4* and *4051-Gal4* in the nerve cord (*Figure 1—figure supplement 2D*): serotonergic A26e neurons (*Huser et al., 2012*; *Okusawa et al., 2014*) (*Figure 1—figure supplement 2E*), and GABAergic A27j neurons (*Fushiki et al., 2016*; *Schneider-Mizell et al., 2016*) (*Figure 1—figure supplement 2F*).

To characterize the morphology of *412-Gal4* segmental interneurons at single cell resolution, we used the 'Flip out' technique (*Basler and Struhl, 1994*; *Wong et al., 2002*). Primary neurites project to the ventromedial neuropil, where they arborize profusely (*Figure 1C–C'*). These medial processes accumulated the dendritic marker, DenMark (*Nicolaï et al., 2010*) (*Figure 1—figure supplement 3A–A'*). A single process emerged from this dendritic region and projected laterally and dorsally back towards the cell body (*Figure 1C'*). These lateral projections accumulated the presynaptic marker Bruchpilot.short$^{mCherry}$ (BRP.short$^{mCherry}$) (*Schmid et al., 2008*) (*Figure 1—figure*

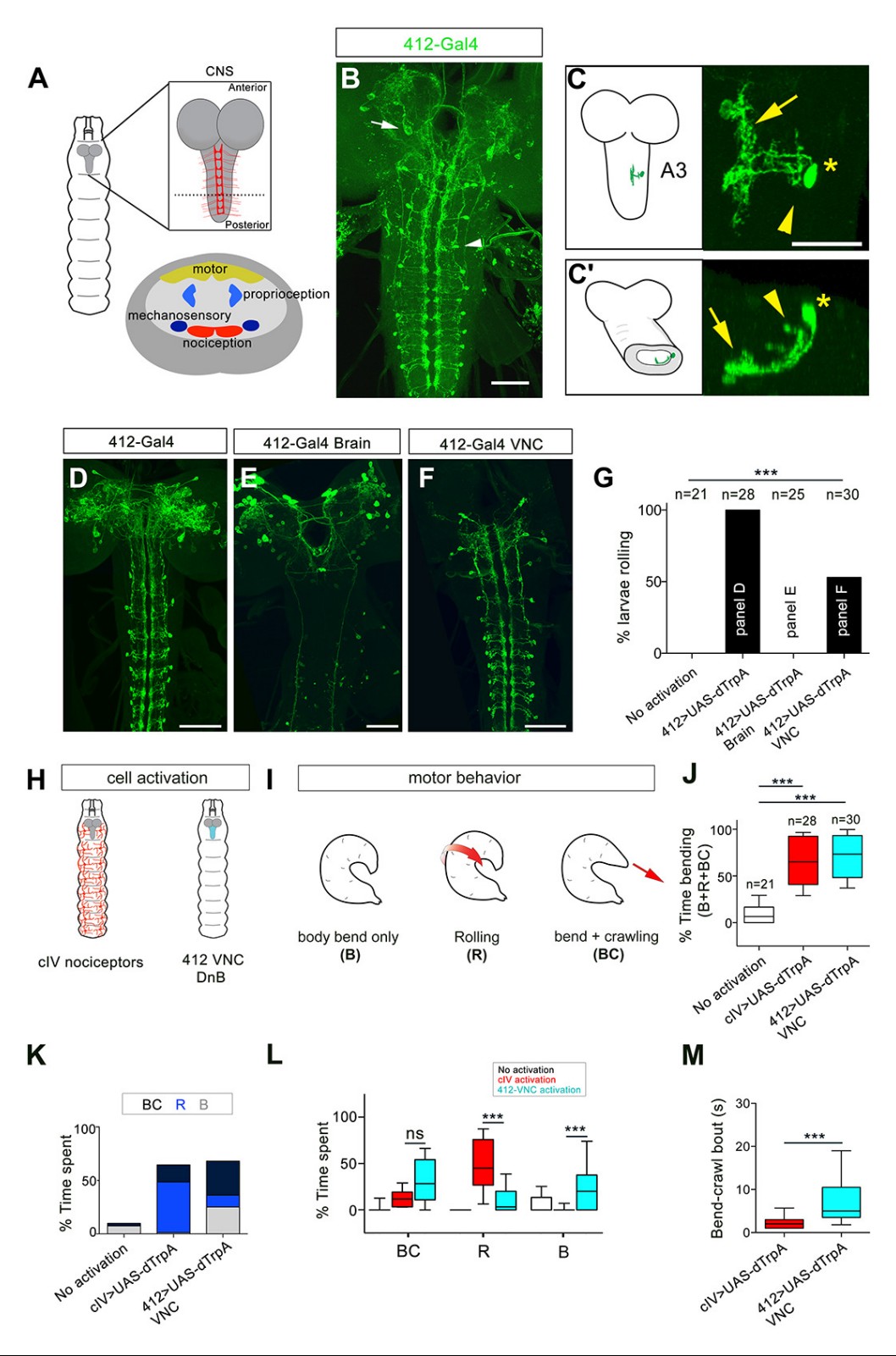

**Figure 1.** Identification of candidate nociceptive interneurons. (**A**) Schematic showing the *Drosophila* larval CNS. Red scaffold represents class IV (cIV) projections. Enlarged transverse section through ventral nerve cord (VNC) is shown below. Color-coded regions depict modality specific locations where sensory axons terminate and motor neuropil. (**B**) *412-Gal4* drives expression in interneurons in the VNC (arrowhead) and neurons in the brain lobes

*Figure 1 continued on next page*

*Figure 1 continued*

(arrow), anti-dsRed, green. (**C–C'**) Dorsal view of the morphology of DnB neuron in segment A3. Medial process is indicated by an arrow and lateral projection by an arrowhead. An asterisk marks the cell body. (**C'**) Transverse section of neuron in C. (**D–F**) Intersectional strategy to target GFP either to D) the brain and VNC, (**E**) brain only or F) VNC only. Green channel shows anti-GFP labeling. (**G**) Percent exhibiting nocifensive rolling during dTrpA1 activation of subsets of *412-Gal4* neurons corresponding to panels D-F. (**H**) Schematic of cIV nociceptors (left) and location of *412-Gal4* VNC neurons (right). (**I**) Schematic of different motor behaviors observed in response to cIV or *412-Gal4* VNC activation. Body bend only, B, larvae entered a curved C-shape but did not roll or crawl; Rolling, R, animals entered C-shape and performed 360° rotations; Bend + crawling, BC, larvae attempted to crawl while remaining in a C-shape. Red arrows show direction of locomotion. (**J**) Total amount of time spent in bent-body positions (B + R + BC) upon dTrpA1-induced activation of cIV neurons and *412-Gal4* VNC neurons. (**K**) Percent of time upon dTrpA1 activation spent in bent-body positions with crawling (black) rolling (blue) or paused (bend-only, gray). (**L**) Percent of time spent during 29 s trial in bent-body positions: bend-crawl, rolling, or bend-only. (**M**) Plot showing length of bend-crawl bouts in seconds upon cIV or *412-Gal4* VNC activation. Box plots show median (middle line) and 25th to 75th percentiles with whiskers representing 10 to 90 percentiles. P values are indicated as *p<0.05, ***p<0.001, as determined by One-way ANOVA with Tukey's multiple comparison's test (**J**), Kruskal-Wallis with Dunn's correction for multiple testing (**L**), or Mann-Whitney (**M**). Scale bars = 50 µm (**B, D–F**), 20 µm (**C**). (See also *Figure 1—figure supplements 1–4*).

DOI: https://doi.org/10.7554/eLife.26016.002

The following source data and figure supplements are available for figure 1:

**Source data 1.** Summary table of graph data and statistical testing for thermogenetic activation experiments.
DOI: https://doi.org/10.7554/eLife.26016.007

**Figure supplement 1.** *4051-Gal4* expression pattern.
DOI: https://doi.org/10.7554/eLife.26016.003

**Figure supplement 2.** Further analysis of *412-Gal4* expression.
DOI: https://doi.org/10.7554/eLife.26016.004

**Figure supplement 3.** DnB polarity analysis.
DOI: https://doi.org/10.7554/eLife.26016.005

**Figure supplement 4.** *412-Gal4, 4051-Gal4,* and off target activation.
DOI: https://doi.org/10.7554/eLife.26016.006

**Figure supplement 4—source data 1.** Summary table of graph data and statistical testing for activation experiments.
DOI: https://doi.org/10.7554/eLife.26016.008

---

supplement 3B–B'). We also observed BRP.short^mCherry accumulation in medial dendrites, suggesting both presynaptic and postsynaptic functions for these arbors (*Figure 1—figure supplement 3B–B'*). Fitting with lineage-based nomenclature, the interneurons labeled by *412-Gal4* were identified as the A09l neurons (*Gerhard et al., 2017*; *Lacin and Truman, 2016*). Because these neurons project 'down' to the ventromedial neuropil, arborize, and sent a reverse projection back towards the cell body, we refer to them as 'Down-and-Back' or DnB neurons.

Next, we assessed the behavioral consequences of activating *412-Gal4* neurons using both thermogenetic and optogenetic approaches. Thermogenetic activation of *412-Gal4* or *4051-Gal4* neurons triggered rolling behavior (*Figure 1—figure supplement 4A–C*; *Video 1*). We observed similar rolling behavior when we activated *412-Gal4* neurons using ReaChR (71% larvae rolling, n = 48) in animals raised with all-*trans*-retinal, an essential co-factor for channelrhodopsin (*Figure 1—figure supplement 4D*; *Video 1*). We next genetically separated *412-Gal4* brain expression from VNC expression by combining the VNC-specific Gal4 inhibitor *tsh-*

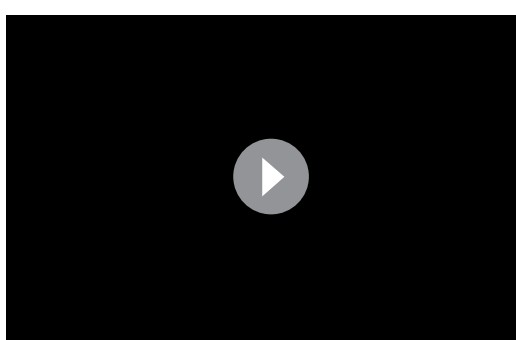

**Video 1.** *412-Gal4* activation induces nocifensive rolling Video shows result of activating *412-Gal4* neurons with dTrpA1 or ReaChR. For ReaChR videos, flashing light indicates 'lights on.'
DOI: https://doi.org/10.7554/eLife.26016.009

*Gal80* with *412-Gal4* (*Figure 1D–G*). Rolling probability was not increased when *412-Gal4* expression was restricted to brain neurons (0% larvae rolling, n = 25; *Figure 1G*). Conversely, to determine whether activity in VNC interneurons can trigger nociceptive rolling, we used an intersectional strategy to drive *Gal4* expression at the intersection of *tsh-LexA* and *412-Gal4* (*412-Gal4* VNC) (*Figure 1F*). Compared to control animals that did not roll (0% larvae rolling, n = 21), activating *412-Gal4* neurons in the VNC, where DnBs reside, increased rolling probability (59% larvae rolling, n = 30; *Figure 1G*). Activation of A26e and A27j, did not significantly increase rolling probability (*Figure 1—figure supplement 4E–G*). These data indicate that activation of DnB neurons in the *412-Gal4* VNC pattern are likely responsible for inducing nociceptive behavior.

## *412-Gal4* VNC interneurons promote nociceptive C-bending and rolling behavior modules

The behaviors induced by *412-Gal4* VNC neuron activation were similar to the rolling behavior generated by noxious stimuli (*Hwang et al., 2007*; *Ohyama et al., 2013*). Class IV (cIV) da neurons function as primary nociceptors in *Drosophila* larvae (*Hwang et al., 2007*). Nocifensive behavior consists of multiple components organized as a sequence, including C-bending, rolling, and escape crawling. To determine whether *412-Gal4* VNC activation induces the same components of the cIV neuron triggered nocifensive response, with the same relative timing, we compared the behavioral consequences of activating cIV neurons to activating DnB neurons (*Figure 1H*). We quantified the behaviors that occur during nocifensive escape by monitoring C-bending without coincident rolling (bend only, B), and bending behavior that coincides with rolling (rolling; R). We also monitored hybrid behaviors, such as bend-crawl (BC) in which crawling larvae were persistently bent (*Figure 1I*). We found that cIV or *412-Gal4* VNC activation led to similar increase in overall time spent in a bent body orientation (B, R, BC) compared to control animals (*Figure 1J*). However, cIV neuron activation more often triggered rolling (R) behavior, whereas *412-Gal4* VNC activation was more likely to induce sustained bending (B or BC) behavior (*Figure 1K–L*; *Video 2*). As an example, activation of *412-Gal4* VNC neurons caused larvae to spend more time performing BC behavior compared to activation of cIV neurons (32% total time; bout mean = 7.8 s vs. 16% total time; bout mean = 2.7 s, respectively; *Figure 1M*). We did not observe 'escape crawl,' or increased forward locomotion speed with minimal turning, upon *412-Gal4* VNC activation. Thus, *412-Gal4* VNC activation can induce C-bending both with and without nociceptive rolling, suggesting modularity in the nocifensive sequence.

While cIV nociceptor activation induces the entire nocifensive escape sequence (C-bend→roll→escape crawl), *412-Gal4* VNC activation can induce rolling or C-bending without rolling, in a modular manner. One possibility is that cIV and DnB neurons can recruit different motor programs based on levels of neural activation. To further examine the behaviors induced by cIV and *412-Gal4* neuron activation, we performed experiments to activate cIV or *412-Gal4* neurons at different intensities. The progression of nocifensive behaviors were monitored over time using frustrated total internal reflection imaging (*Risse et al., 2013*). We opto-genetically activated DnB or cIV neurons using 525 nm LED at four different light intensities, Lowest (~45 lx), Low (~200 lx), Moderate (~850 lx), Highest (~1450 lx), for 10 s by expressing *UAS-ReaChR*, under the control of *412-Gal4*, or *PPK*[1.9]*-Gal4*, respectively. Again, we monitored bending only (B), rolling (R), along with crawling (C), and pausing (P). Bend-Crawl (BC) was not observed in this experimental paradigm during activation of either population of neurons. Behavioral ethograms and bending analyses showed that when cIV neurons were activated, animals typically showed sharp increases in bending coupled with rolling within 1 s of activation across all activation intensities (*Figure 2A,B,E*). Conversely, *412-Gal4* activation led to persistent bending increases, which only coincided with periods of rolling within 10 s at Moderate levels of

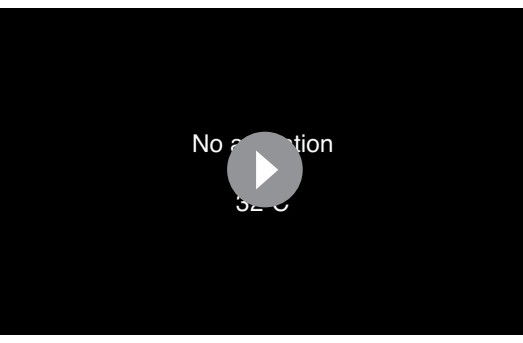

**Video 2.** Activating *412-Gal4* neurons in the VNC causes body bending Video shows result of activating *412-Gal4* neurons exclusively in the brain or the VNC, compared to cIV activation
DOI: https://doi.org/10.7554/eLife.26016.010

activation, and within 5–10 s at the Highest intensity (*Figure 2C–E*). These data suggest that cIV activation triggers rolling in concise bouts, while *412-Gal4* activation initially elicits bending, which may or may not progress into rolling events, depending on the intensity of activation.

Together, these data suggest that whether *412-Gal4* activation of DnB neurons trigger bending *vs.* rolling modules is dose-dependent, as low levels of activation induce persistent bending, and higher levels of activation trigger bending followed by rolling. Activation of bending independently of rolling suggests divergence in the downstream neural circuitry underlying nocifensive escape behavior.

## DnB neurons function downstream of cIV neurons

We next performed behavioral and physiological analysis to determine whether DnB neurons functioned downstream of cIV nociceptors. We first tested whether DnB neurons respond to noxious stimuli by performing calcium imaging experiments. We drove expression of GCaMP6m (*Chen et al., 2013*) in DnB neurons using *412-Gal4* in a partially dissected preparation and applied a local ramped heat stimulus to abdominal segments. We observed increased GCaMP6m fluorescence in DnB neurons (identified by their morphology) beginning at 39°C and plateauing at approximately 42°C (*Figure 3A–C*; *Video 3*), fitting well with prior studies showing cIV neuron spiking above 38°C (*Terada et al., 2016*; *Tracey et al., 2003*; *Xiang et al., 2010*). Silencing cIV neurons reduced calcium responses by 68% during noxious stimulation and delayed the onset of the calcium response (*Figure 3D–F*; *Video 3*). Taken together, these data support a role for DnB neurons in the transduction of noxious heat stimuli from cIV sensory neurons.

We next tested whether *412-Gal4* neurons induce nociceptive behavior downstream of cIVs. Silencing cIV neurons by driving tetanus toxin light chain (TNT) (*Karuppudurai et al., 2014*) under the control of a cIV-specific driver *R38A10-LexA* reduced rolling behavior in a local nociceptive heat assay (*Figure 3G–H*). Activation of DnBs using *412-Gal4* largely bypassed this inhibition and induced rolling in 82% of animals (*Figure 3I*). Thus, these data suggest that DnB neurons act functionally downstream of cIV activity.

## DnB interneurons are required for robust nociceptive rolling and C-bending

Since our data show that *412-Gal4* activation is sufficient to trigger bending and rolling, we next performed silencing experiments to ask whether DnB neurons are required for bending and rolling during nocifensive escape. To test the requirement for DnBs in nociceptive behavior, we took an intersectional approach to further refine our manipulations. We first screened the Fly Light Gal4 database of approximately 7000 enhancer-based Gal4 expression patterns (*Jenett et al., 2012*; *Li et al., 2014*). We identified several lines with broad expression in the VNC and performed a secondary screen on corresponding LexA versions by crossing them to *412-Gal4, 8X-lexAop2FLPL,* and *10XUAS > stop > myr:GFP* (*Shirangi et al., 2013*). This approach led to labeling at the intersection of the LexA and Gal4 lines. We identified one line, *R70F01-LexA,* that supported intersectional expression in abdominal DnB neurons, weakly in a small number of other VNC neurons, including A27j neurons, and only rarely in one brain neuron (*Figure 4—figure supplement 1A–B'*). We used the *R70F01-LexA∩412-Gal4* (*R70F01∩412*) strategy to drive expression of Kir$^{2.1}$-GFP (*Shirangi et al., 2013*), a hyperpolarizing channel (*Baines et al., 2001*) (*Figure 4A*). As has been described (*Shirangi et al., 2013*), we observed all-or-none expression of Kir$^{2.1}$-GFP when larvae were visualized after experiments to assess *Kir$^{2.1}$-GFP* expression (32%, n = 125). Animals were classified as 'non-silenced' (i.e. lacking Kir$^{2.1}$-GFP expression) controls or '*R70F01∩412*-silenced' (i.e. with Kir$^{2.1}$-GFP expression in VNC).

Upon exposure to a noxious heat surface (40°C), control animals showed a typical nociceptive sequence of (1) C-shaped body bending and rolling, (2) brief forward crawling with lateral bending and occasional rolling (transition), and (3) rapid forward escape crawling (*Figure 4B*; *Video 4*). During rapid forward crawling we observed no C-bending or rolling (*Video 4*). Since activation of *412-Gal4* VNC, which includes DnBs, increased both C-bending, and rolling, we first asked whether rolling was affected in our global heat assay upon DnB silencing, using the *R70F01∩412* strategy. We found that silencing *R70F01∩412* neurons did not abolish rolling, but significantly reduced the absolute number of rolls per trial (rolling median = 0, *R70F01∩412* silencing; median = 3, control groups;

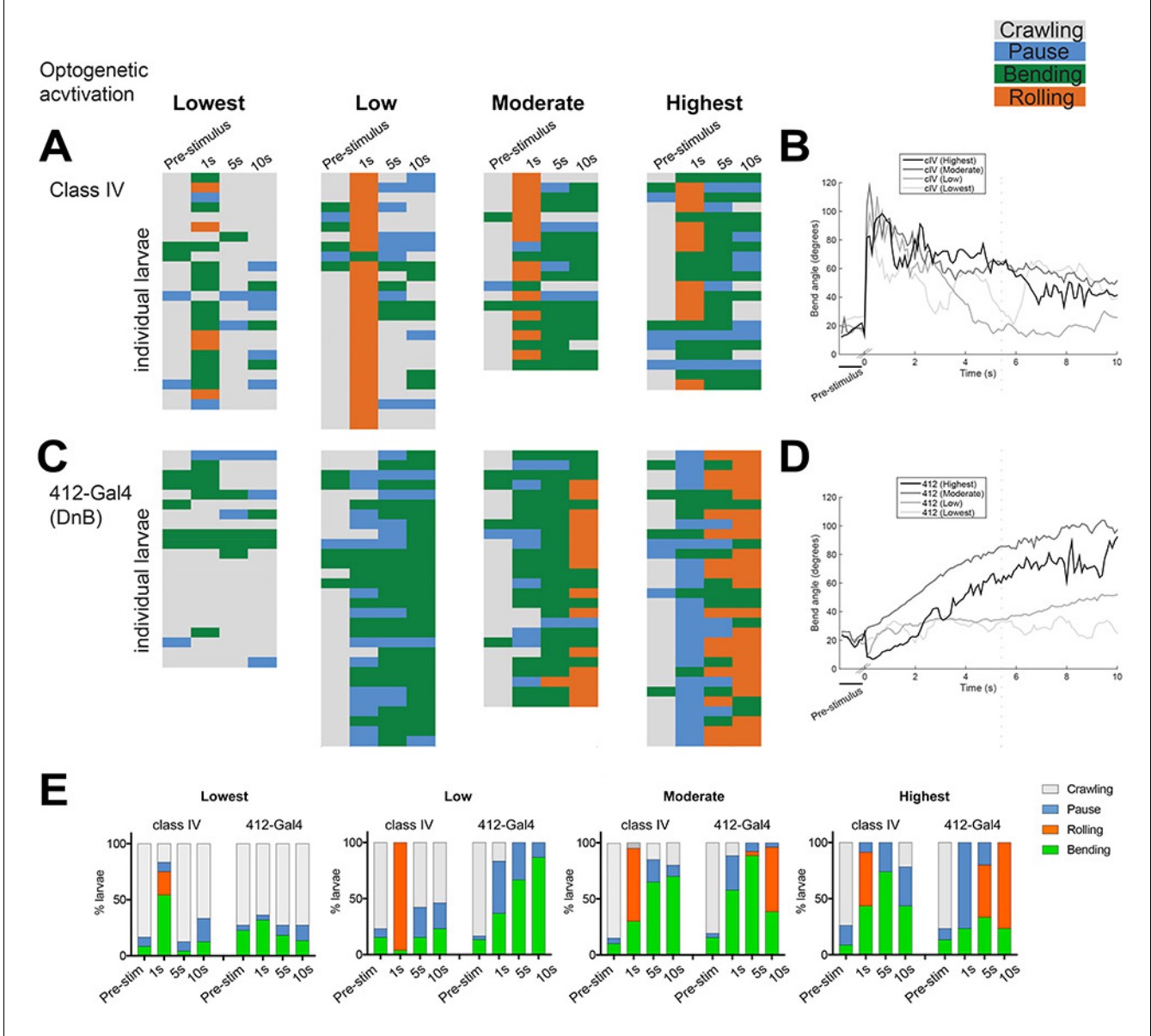

**Figure 2.** DnBs promote bending and rolling in a dose-dependent manner. (**A, C**) Behavior ethograms upon optogenetic stimulation of *412-Gal4* or class IV neurons. Groups of animals expressing ReaChR in either population were subjected to optogenetic activation at different light intensities for 10 s: Lowest (~45 Lux), Low (~200. Lux), Moderate (~850 Lux) and Highest (~1450 Lux). Behavior events are color-coded: crawling (grey), pause (blue), bending (green), and rolling (orange). (**A**) Behaviors triggered upon optogenetic activation of class IV neurons. Lowest, n = 24. Low, n = 26 Moderate, n = 20; Highest, n = 23. (**B**) Behaviors triggered upon optogenetic activation of *412-Gal4* neurons. Lowest, n = 22. Low, n = 30; Moderate, n = 26; Highest, n = 30. (**C**) Percent of larvae exhibiting crawling (grey), pausing (blue), rolling (orange) and bending (green) across different activation intensities. (See also *Figure 2E*).

DOI: https://doi.org/10.7554/eLife.26016.011

The following source data is available for figure 2:

**Source data 1.** Summary table of behavioral responses to dose-dependent optogenetic activation.
DOI: https://doi.org/10.7554/eLife.26016.012

*Figure 4C*), without affecting the order of the rolling bout in the nociceptive sequence, or the latency to initiate the first roll (*Figure 4—figure supplement 1C*). *R70F01∩412*-silenced animals took more time to complete a roll (mean = 1.54 s) compared to control animals (mean = 0.84 s), indicating that *R70F01∩412* neurons are important for rapid nociceptive rolling behavior (*Figure 4D*).

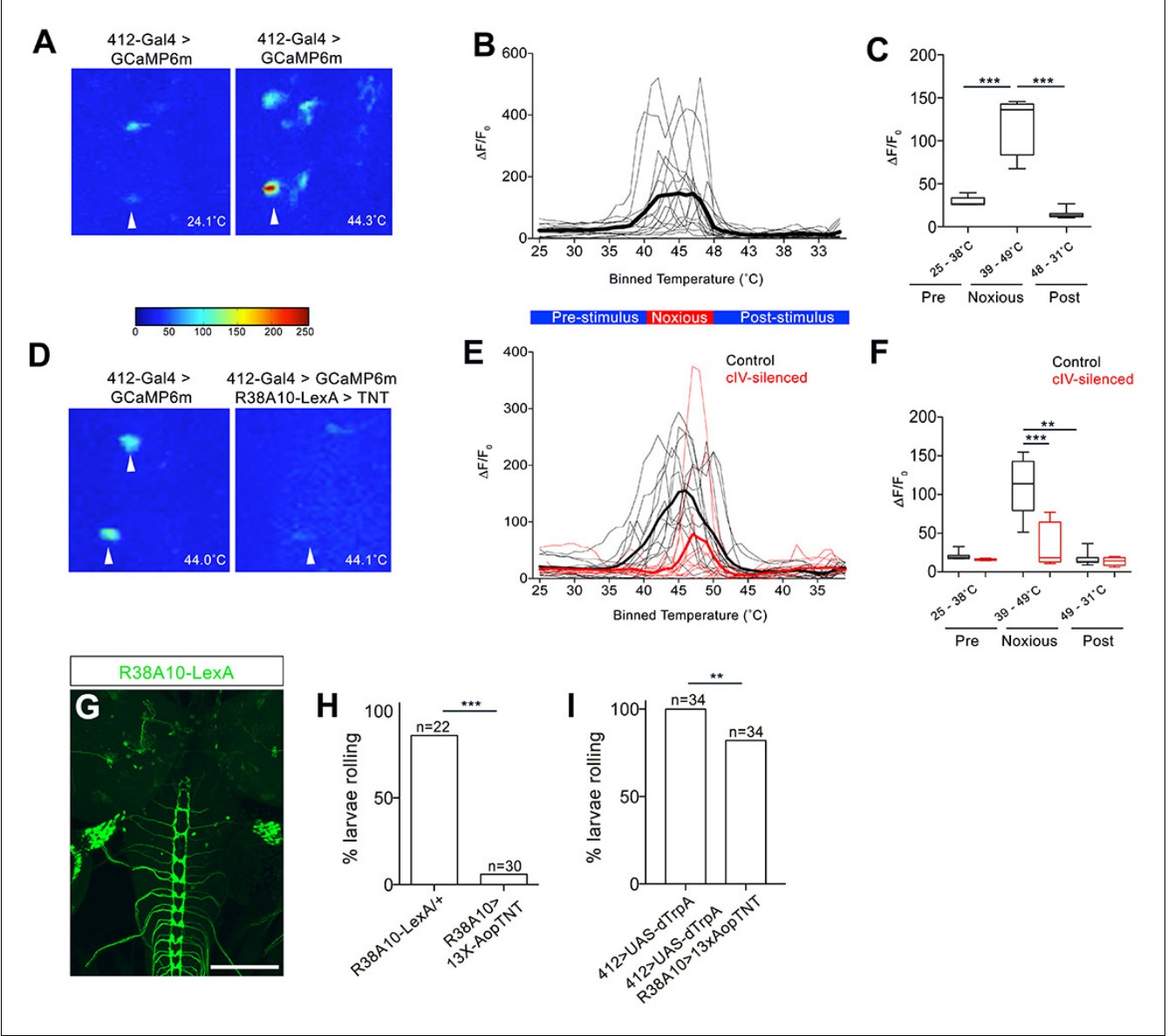

**Figure 3.** DnBs are activated by noxious heat downstream cIV sensory neurons. (**A**) Representative heat maps showing $Ca^{2+}$ responses in DnB cell bodies (arrowhead) before (~24°C) and during (~44°C) local noxious heat stimulation of the body wall. (**B**) Individual $Ca^{2+}$ responses (thin lines) and average of all trials (bold) represented as $\Delta F/F_0$ in DnB cell bodies (n = 15). Larvae received local heat stimulation at segment A7 using a heat probe that was increased from ~24–49°C, then cooled to ~30°C. (**C**) GCaMP signal binned for 25–38°C (below noxious threshold), 39–49°C (above noxious threshold), and 48–31°C (post-stimulus cool down). (**D**) Representative heat maps showing $Ca^{2+}$ responses in DnB cell bodies at ~44°C (arrowhead) with or without cIVs silenced with *R38A10-LexA* driving TNT. (**E**) Individual $Ca^{2+}$ responses (thin lines) and average of all trials (bold) in DnB cell bodies during heating and cooling, Black lines represent control (n = 12) and red lines represent cIV silenced trials (n = 11). (**F**) GCaMP signal binned for 25–38°C (below noxious threshold), 39–49°C (above noxious threshold), and 49–31°C (post-stimulus cooling) for control and cIV silenced trials. (**G**) *R38A10-LexA* driven *13XlexAop2-IVS-myr::GFP* labels cIV sensory neurons (anti-GFP, green) and sparse labeling of brain neurons. (**H**) Percent of larvae rolling in response to local noxious stimuli decreased when cIV neurons were silenced using *R38A10-LexA* to drive tetanus toxin light chain (TNT). (**I**) Percentage of animals exhibiting rolling responses when *412-Gal4* neurons were induced by dTrpA1 with and without cIV-silencing via *R38A10-LexA* driven tetanus toxin light chain (TNT). *p-value<0.05, **p-value<0.01, ***p- value <0.001 by Chi squared test with Bonferroni correction for cases of multiple testing (**H–I**). Kruskal-Wallis with Dunn's correction for multiple testing and post-hoc Unpaired T-test or Wilcoxon test (**C**) or post-hoc T-test or Wilcoxon test (**F**). Scale bars = 70 µm (**G**).

DOI: https://doi.org/10.7554/eLife.26016.013

The following source data is available for figure 3:

**Source data 1.** Summary table of graph data and statistical testing for functional imaging and nociceptive experiments.
Genotypes, number of animals tested, graph data and statistical testing presented for DnB GCaMP imaging and nociceptive behavior experiments.
DOI: https://doi.org/10.7554/eLife.26016.014

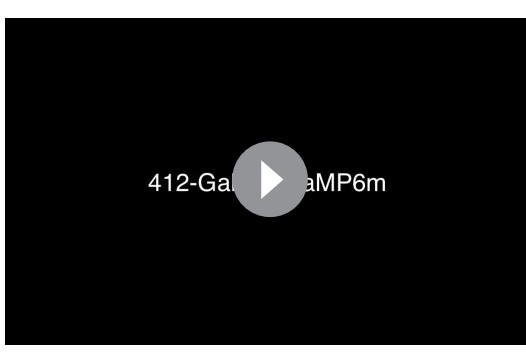

**Video 3.** DnB neurons are activated by noxious thermal stimuli Video shows GCamp6m fluorescence in the VNC of a partially dissected larvae, with and without class IV neural activity. Arrows used to indicate cell body, axon, and dendrites.
DOI: https://doi.org/10.7554/eLife.26016.015

To determine whether silencing A27j neurons contributed to the reduced rolling observed in the *R70F01∩412* silencing experiments, we expressed Kir[2.1] in A27j neurons using *R38H01-Gal4* (*Schneider-Mizell et al., 2016*). We found that silencing A27j neurons did not significantly reduce the number of rolls during global noxious stimulation (*Figure 4—figure supplement 1D*). Thus, the reduction of nociceptive behavior observed during *R70F01∩412* silencing appears to be a consequence of reducing DnB function.

Given the *412-Gal4* VNC activation data suggesting that DnB neurons also promote C-bending, independent of rolling, we asked how bending behavior during nocifensive escape is affected by the reduction of DnB activity. First, we analyzed the amount of time spent bending vs. rolling in *R70F01∩412*-silenced animals (*Figure 4—figure supplement 1E*). We found a significant decrease in the percentage of time that *R70F01∩412*-silenced larvae exhibited bend-rolling behavior. Bend-roll behaviors might have been replaced by bend-crawling bouts, as we found a modest increase in the time that larvae spent in the bend-crawl mode (*Figure 4—figure supplement 1E*). This conclusion is supported by the finding that *R70F01∩412* silencing led to an increase in bending without rolling compared to control larvae (49% vs. 14%) (*Figure 4—figure supplement 1F*). These results suggest that reduced rolling probability upon DnB silencing coincides with an increase in bending events that do not result in rolling, and thus an abbreviated nocifensive sequence.

Given the preferential activation of bending behavior upon *412-Gal4* activation, we considered whether *R70F01∩412* silenced larvae show deficits in the acquisition of the C-bend. To quantify curvature along the larval body during nocifensive escape, we adapted a technique used to visualize curvature during slime mold migration (*Driscoll et al., 2011*; *Driscoll et al., 2012*). This technique allows for quantification and visual representation of curvature along the larval body. Briefly, 300 points were distributed along the boundary of the larval body (*Figure 4E*). A curvature index (CI) was calculated at each point, and color-coded as a heat map of CI values (*Figure 4E*), assigning deeper C-bends with higher CI values. We focused our analysis on curvature along the inner C-bend, and plotted changes in local bending over time as kymographs (*Figure 4F–G*) to visualize the evolution of C-bends over a nocifensive escape bout. During rolls (360° rotations) or attempted rolls (<360° rotations), *R70F01∩412*-silenced larvae displayed lower curvature index values (low CI values in blue-yellow range) along the inner C-shape compared to non-silenced animals (high CI values in the orange-red range) (*Figure 4F–H*). To quantify the difference in CI distribution between groups, we categorized CI values into either 'High,' or 'Low' curvature (see methods) and plotted the percent of values that fell within each category per animal. When compared to non-silenced animals, *R70F01∩412*-silenced animals had a significantly higher percentage of points in the Low curvature range, and fewer in the High curvature range, during rolling, or 'attempted' rolling events (*Figure 4I*). Together, these data further support a role for DnB neurons in generating C-bending and rapid rolling. Moreover, rolling probability or efficiency may be related the extent of body curvature.

To exclude non-specific motor defects caused by DnB suppression, we examined crawling behavior while silencing DnB neurons with *412-Gal4* driven TNT (*Sweeney et al., 1995*). We found that aside from a modest increase in crawling speed, crawling was intact, suggesting that DnB neurons play a specific role in escape motor circuitry (*Figure 4—figure supplement 1G*).

## DnB neurons receive synaptic input from nociceptive and gentle-touch neurons

We next utilized an electron microscopic (EM) volume of the first instar larval CNS (*Ohyama et al., 2015*) to reconstruct DnB upstream connections (*Schneider-Mizell et al., 2016*). We reconstructed

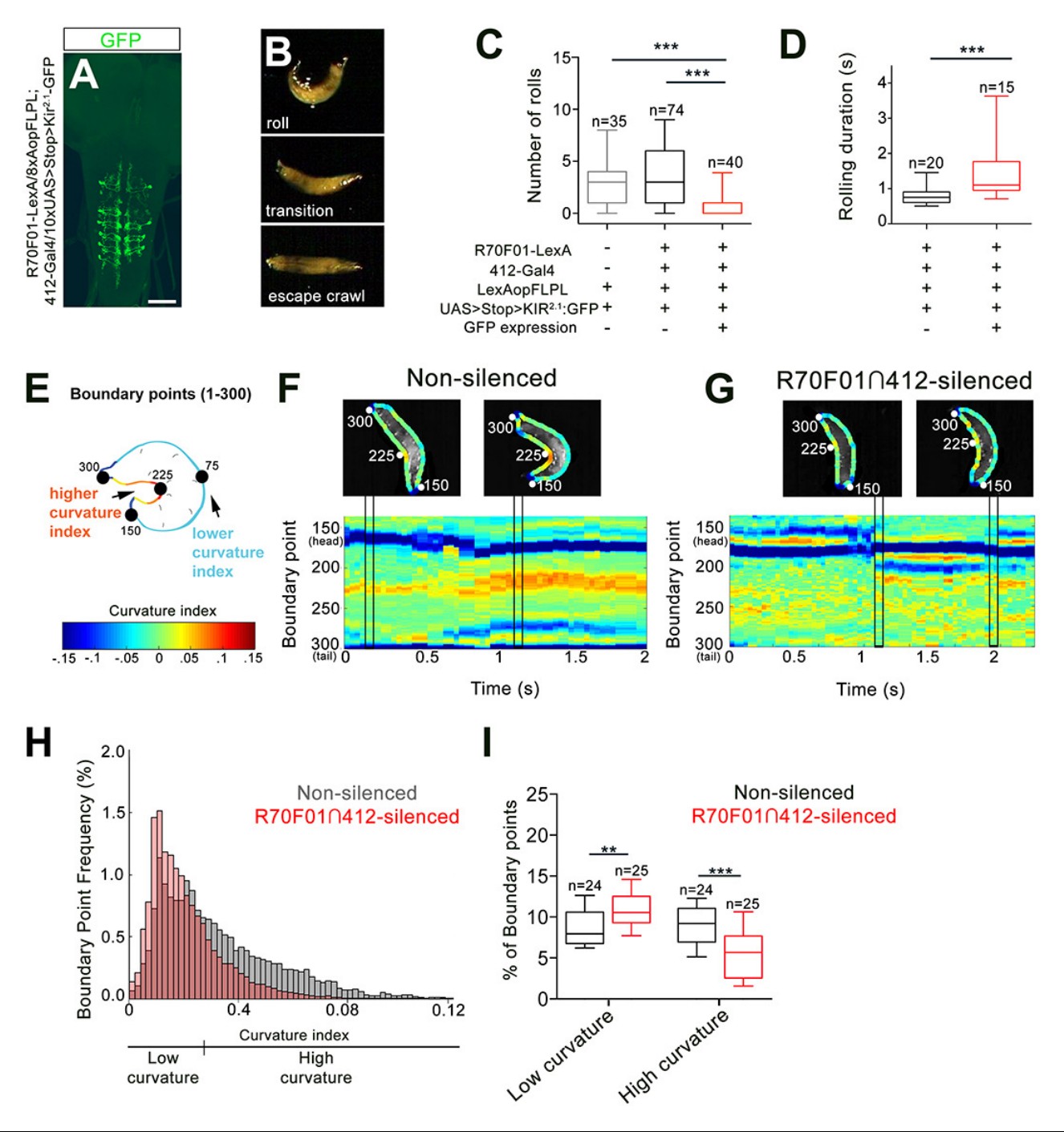

**Figure 4.** DnBs are required for body bending during nocifensive rolling. (**A**) Labeling DnB neurons using *R70F01-LexA* driving *8X-Aop2-FLPL*, and *412-Gal4* driving *10XUAS > Stop > Kir²·¹-GFP* (anti-GFP, green). (**B**) Global heat stimulus leads to rolling (top), transition period (middle), and an increase in forward crawling speed (escape crawl; bottom). (**C**) Number of rolls per trial. 'Non-silenced' animals lacked *Kir²·¹-GFP* expression and '*R70F01∩412*-silenced' animals exhibited GFP expression. (**D**) Rolling duration of the 1st roll for animals that completed 360° rotations. (**E**) Schematic of larva with curvature analysis. Program outlines boundary of larval body and assigns a curvature index value at each of 300 boundary points. The curvature values are represented as a heat map along the larval body. (**F**) Representative kymograph showing curvature indices (CI) along C-bend (spanning boundary points 150 and 300) in non-silenced animals during the duration of the first roll. Larval images above kymographs represent CI at each boundary point position along the outline of the entire body at time points when the animal acquires a low curvature (left) or high curvature shape (right; indicated in plots as vertical tracks). (**G**) Representative kymograph showing curvature indices along C-bend in *R70F01∩412*-silenced animals. Kymograph is as represented in (**F**). (**H**) Frequency distribution of concave curvature indices (CI) of all boundary points across the bending duration for animals that rolled (360° turn) and 'attempted' rolling (*i.e.* 0–360° rotations) separated into low curvature (CI <0.027) and high curvature (CI >0.027) values. (**I**) Percentage of

*Figure 4 continued on next page*

*Figure 4 continued*

boundary points that fall into the category of low curvature (CI <0.027) and high curvature (CI >0.027) values. Scale bar = 50 μm (**A**). Box plots show median (middle line) and 25th to 75th percentiles with whiskers representing 10 to 90 percentiles. P values are indicated as *p<0.05, **p<0.01 ***p<0.001, as tested by Kruskal Wallis with Dunn's correction followed by post hoc Mann Whitney (**C**), Mann-Whitney (**D**), or MANOVA with bonferroni correction, followed by posthoc unpaired T-test (**I**). (See also ***Figure 4—figure supplement 1***).

DOI: https://doi.org/10.7554/eLife.26016.016

The following source data and figure supplements are available for figure 4:

**Source data 1.** Summary table of graph data and statistical testing for silencing experiments.
Genotypes, number of animals tested, graph data and statistical testing presented for R70F01∩412 silencing experiments and curvature analysis.
DOI: https://doi.org/10.7554/eLife.26016.018

**Figure supplement 1.** Intersectional labeling strategy, effect of silencing A27j neurons and effect of silencing *412-Gal4* neurons on somatosensory behavior.
DOI: https://doi.org/10.7554/eLife.26016.017

**Figure supplement 1—source data 1.** Summary table of graph data and statistical testing for silencing experiments.
Genotypes, number of animals tested, graph data and statistical testing presented for silencing experiments on nociceptive, gentle-touch and crawling assays.
DOI: https://doi.org/10.7554/eLife.26016.019

bilaterally symmetric DnB neurons in segment A1 (***Figure 5A***)(***Gerhard et al., 2017***). DnB neurons in A1 receive 45.5% of their input from cIV neurons, consistent with a role in nociceptive behavior. Additional inputs come primarily from other sensory neuron subtypes. Mechanosensitive cIII neurons showed 15% cumulative input, class II neurons showed 4% cumulative input, and external sensory (es) neurons showed 3% cumulative input (***Figure 5B–D***). Inputs to DnB neurons were primarily sensory, with the sole non-sensory input provided by the putative local inhibitory handle-A neurons (***Jovanic et al., 2016***)(***Figure 5B***). These findings suggest that DnB neurons integrate inputs from multiple sensory modalities, with dominant input from cIV nociceptors.

We next examined the localization of sensory inputs along the DnB neuron. EM data showed that the axons of cIII, cIV, and es neurons all provide input onto DnB dendrites (***Figure 5—figure supplement 1A***). cII axons are distinguished from other somatosensory axon projections by their short collateral branches (***Grueber et al., 2007***), which led to the proposal that they might connect with divergent downstream circuits. Interestingly, the input to DnBs from cII neurons came entirely from these collateral axon branches onto the lateral-most DnB axons (***Figure 5—figure supplement 1A***), raising the possibility of presynaptic modulation of DnBs by cII neurons.

cIII and cIV axons terminate in distinct adjacent areas of the neuropil (***Grueber et al., 2007***), implying that they synapse on different regions of the DnB dendrite (***Figure 5—figure supplement 1B–B''***). Indeed, co-labeling DnBs and cIV axons revealed overlap between DnB medial dendrites and cIV axon terminals (***Figure 5—figure supplement 1C–C''***). A lateral domain of the DnB dendritic field did not overlap with the cIV terminals, but did overlap with cIII axons (***Figure 5—figure supplement 1D–D''***) labeled by *nompC-LexA* (***Shearin et al., 2013***). Our EM and anatomical studies therefore reveal that three distinct sensory inputs target DnB neurons in a spatially segregated manner on dendrites and axons.

Notably, activation of DnBs using *412-Gal4* across different intensities did not result in gentle-touch responses (i.e. recoil, backward crawl, head turns) (***Kernan et al., 1994***), but did produce an increase in pausing, which can be either a gentle-touch or light response (***Figure 2***) (***Kernan et al., 1994***; ***Lacin and Truman, 2016***). Moreover, silencing *412-Gal4* neurons using TNT (***Sweeney et al., 1995***) did not disrupt responses to gentle touch (***Figure 4—figure supplement 1H–I***). Thus, despite connections with gentle touch sensory neurons, DnB neurons do not

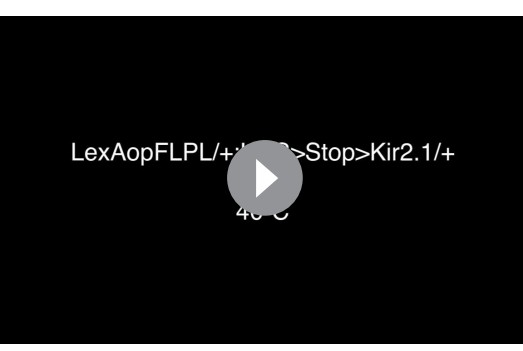

**Video 4.** Silencing R70F01∩412 neurons reduces body curvature during rolling Video shows defective rolling upon R70F01∩412 neuron silencing
DOI: https://doi.org/10.7554/eLife.26016.020

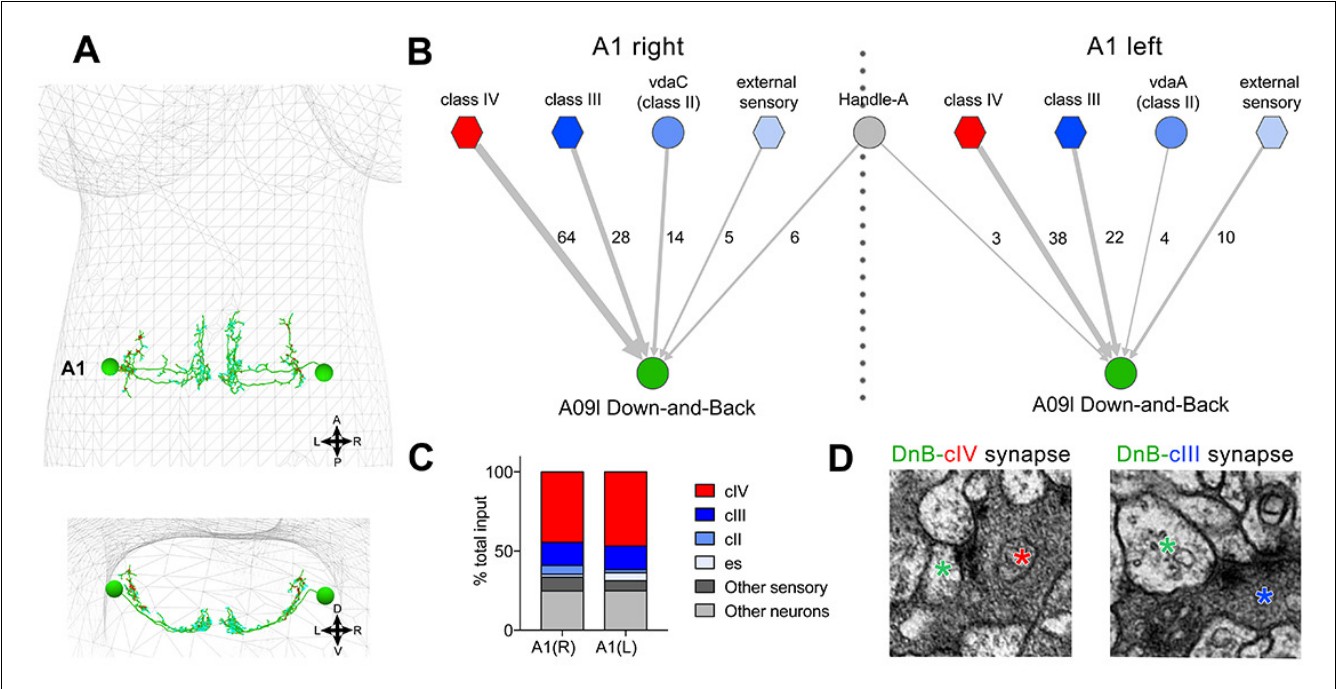

**Figure 5.** Connectome of sensory and interneuron inputs to DnB neurons. (A) First instar larval brain with bilateral reconstruction of DnB neuron morphology in segment A1. Cyan and red dots indicate input and output synapses, respectively. Top, dorsal view; bottom, transverse view. (B) Connectome of inputs onto DnB neurons in right and left A1 hemisegments. Numbers of synaptic connections between segment A1 neurons in top row and DnB neurons are shown. Width of arrow corresponds to degree of synaptic connectivity. Circles represent individual neurons, and hexagons represent groups of neurons. (C) Percent of input provided to total postsynaptic sites on right and left A1 DnB as a function of cell class (not restricted to segment A1). cIV nociceptors provide dominant input to DnBs. (D) Electron micrographs of DnB-cIV and DnB-cIII synapses. (See also *Figure 5—figure supplement 1*).

DOI: https://doi.org/10.7554/eLife.26016.021

The following source data and figure supplement are available for figure 5:

**Source data 1.**

DOI: https://doi.org/10.7554/eLife.26016.023

**Figure supplement 1.** Down-and-Back neurons receive spatially segregated sensory input.

DOI: https://doi.org/10.7554/eLife.26016.022

appear to play a major role in gentle-touch responses.

## EM reconstruction reveals direct connections to premotor neurons and nociceptive integrators

To gain insights into circuit mechanisms underlying nociceptive motor behaviors we performed EM reconstruction of downstream partners of DnB neurons. We identified the complete set of neurons that receive DnB synaptic input using this approach (*Ohyama et al., 2015*; *Schneider-Mizell et al., 2016*) (*Figure 6A,D*). The neurons with the highest numerical connection with DnB neurons (>3 synapses; *Figure 6B*) could be broadly divided into two groups: 'nociceptive integrators' (*Figure 6A,C*) and premotor neurons (*Figure 6D–E*). Nociceptive integrators, exemplified by TePn05 (*Gerhard et al., 2017*), are nodes of convergence of multiple nociceptive neuron types (*Figure 6A, C*). TePn05 makes ascending projections along the nerve cord that are postsynaptic to DnBs and cIV sensory neurons, and that are presynaptic to Basin-2, 4 nociceptive interneurons (*Figure 6A*) (*Ohyama et al., 2015*). TePn05 thus provides a path for communication between DnB and Basin circuits.

DnB connections with premotor neurons provide a potential route to drive escape behavior, particularly the robust C-bending. DnBs form the most synapses with segmentally repeating premotor neurons: A27k, A01d-3, A02g and A02e, (6–18 synapses/hemisegment) (*Figure 6D–E*). With one exception, synapses are made on the ipsilateral side of the nerve cord (*Figure 6E*). A01d-3

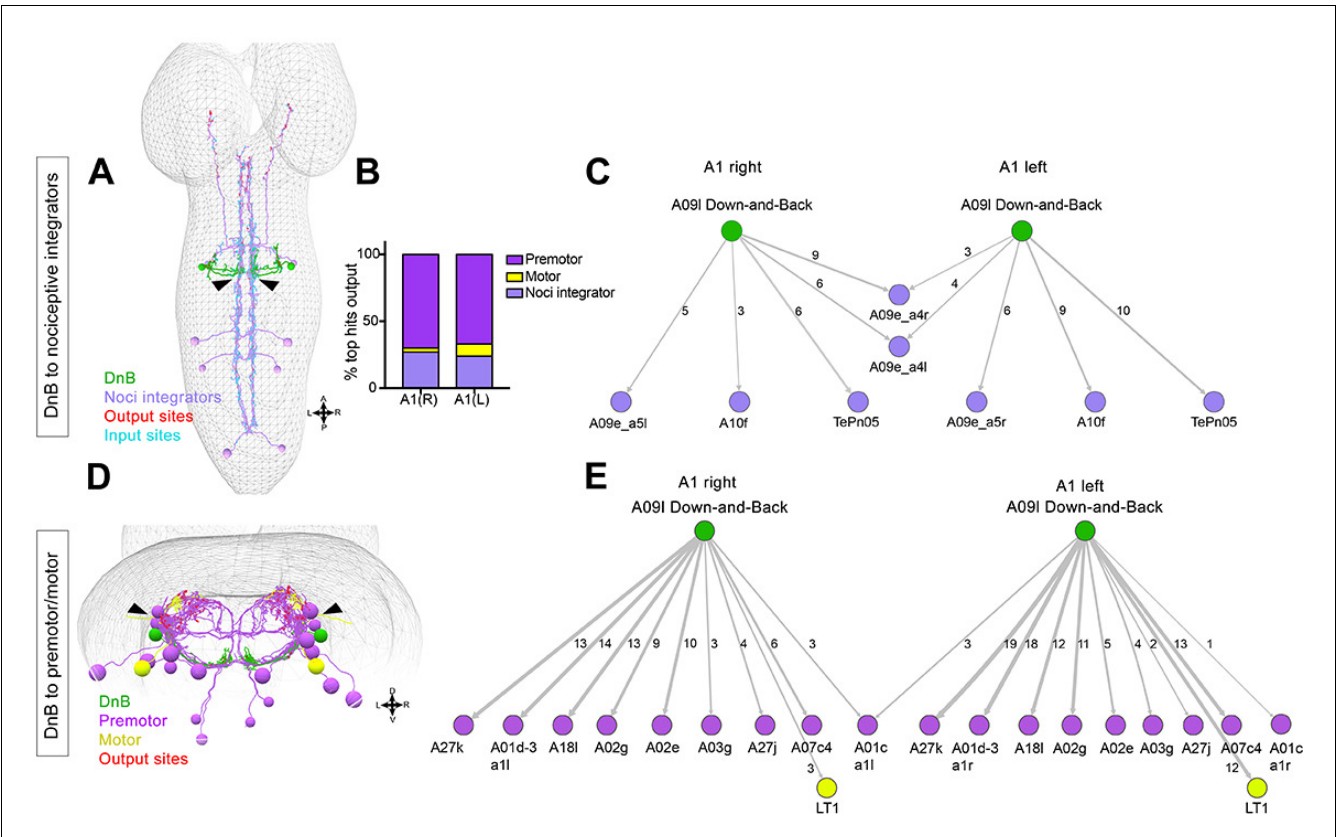

**Figure 6.** Connectome of DnB to premotor and nociceptive interneuron outputs. (**A**) First instar larval CNS showing reconstruction of DnB neurons (green), and nociceptive integrating interneurons (purple). Output synapses are indicated in red and input synapses in cyan. Nociceptive interneurons primarily receive input from output sites on DnB dendrites. (**B**) Percent of top hits' (>3 synapses) output from right (DnB a1R) and left (DnB a1L) A1 DnB neurons as a function of cell type. Premotor circuits and nociceptive integrators are dominant outputs of DnB neurons. (**C**) Identities of nociceptive integrating targets for right and left DnB neurons in A1. Numbers of synapses reconstructed are indicated. Width of arrow corresponds to degree of synaptic connectivity. (**D**) First instar larval CNS showing reconstruction of DnB neurons (green), premotor (purple), and motor targets (yellow). Premotor neuron output synapses (red dots) located primarily in motor domain (arrowhead). (**E**) Identities of premotor targets for right and left DnB neurons in A1. Numbers of synapses reconstructed are indicated. Dominant outputs are A27k and A01d-3 premotor neurons. Width of arrow corresponds to degree of synaptic connectivity. (See also *Figure 6—figure supplement 1*).

DOI: https://doi.org/10.7554/eLife.26016.024

The following source data and figure supplement are available for figure 6:

**Source data 1.** Summary table for output connectivity graph.
Percentage of top hit neurons (>3 synapses with DnB) that fall into the category: premotor, motor, or nociceptive integrator neurons.
DOI: https://doi.org/10.7554/eLife.26016.026

**Figure supplement 1.** Additional properties of Down-and-Back output signaling.
DOI: https://doi.org/10.7554/eLife.26016.025

interneurons receive input from contralateral DnB neurons (*Figure 6E*), and project to interneurons in contralateral posterior segments. Some premotor neurons downstream of DnB are implicated in duration and propagation of segmental waves during larval forward locomotion, including A02g and A02e (part of *period*-positive median segmental interneuron, or PMSI, inhibitory interneurons), (*Kohsaka et al., 2014*), and A27k (*Fushiki et al., 2016*; *Zwart et al., 2016*). DnBs also make modest connections with motor neurons innervating muscle LT1 (*Zwart et al., 2016*)(*Figure 5B*). The connections of DnB to multiple premotor neurons could promote changes in body curvature during nociceptive escape behavior.

Notably, DnB output synapses to these different groups of downstream neurons are anatomically segregated. Nociceptive integrators receive input from DnB presynaptic sites located on the medial

DnB dendrite, whereas premotor neurons receive synaptic inputs from lateral DnB axons (*Figure 6A, D* arrowheads).

We additionally used immunolabeling and EM to examine the transmitter identity of DnB neurons. First, we introduced *cha$^{3.3kb}$-Gal80* (*Kitamoto, 2002*), which expresses in a large subset of excitatory cholinergic neurons, into *412-Gal4, UAS-mCD8:GFP* animals. We found a reduction in GFP signal in both cell bodies and in medial processes of DnB neurons (*Figure 6—figure supplement 1A–A'*). Next, we labeled cholinergic neurons with an antibody against choline acetyltransferase (ChAT) and observed co-localization between DnB neurons and ChAT, but not with vGLUT or GABA (*Figure 6—figure supplement 1B–D''*). We confirmed a cholinergic identity by co-labeling *412-Gal4* and ChAT tagged with eGFP (*Nagarkar-Jaiswal et al., 2015*). We observed ChAT-eGFP expression in DnB cell bodies and axonal processes (*Figure 6—figure supplement 1E–E'''*). Together these data support a cholinergic identity of DnB interneurons.

We noted in EM sections that in addition to small clear vesicles characteristic of acetylcholine release, large dense core vesicles accumulate at many DnB presynaptic sites, which is indicative of aminergic or peptidergic signaling (*Figure 6—figure supplement 1F*). Thus, in addition to affecting nociceptive circuitry by direct synaptic connections, DnB neurons may modulate circuitry through neuropeptidergic or aminergic signaling. Consistent with this possibility, we found that PreproANF fused with emerald GFP, which accumulates at peptidergic output sites (*Rao et al., 2001*), overlaps with DnB axons, suggesting that DnBs possess the machinery to package and release neuropeptides (*Figure 6—figure supplement 1G*).

## Premotor neurons and command neurons acting downstream of DnBs

Our EM reconstruction revealed that DnB neurons had the highest number of synaptic connections with premotor neurons. A02e and A02g belong to the PMSI group (*Kohsaka et al., 2014*). To examine roles for PMSI neurons in nociceptive behavior, we silenced this group of cells using *period-Gal4* and monitored rolling behavior on the global heat assay. We found that silencing of *period-Gal4* neurons significantly reduced the number of rolls per trial without significantly affecting the rolling bout length or the percent of animals exhibiting bending and rolling. (*Figure 7—figure supplement 1A–D*). We note that PMSIs also include A02a-j neurons (*Kaneko et al., 2017*), so although our manipulations are not specific to A02e and A02g neurons, these results are consistent with PMSI neurons, perhaps including those downstream of DnBs, promoting robust nocifensive behavior.

The direct pathways from DnB neurons to nociceptive integrators (*Figure 6C*; *Figure 7A*) provide a possible functional link with Goro neurons, command-like neurons for rolling (*Ohyama et al., 2015*). Goros receive indirect input from Basin nociceptive interneurons to promote rolling behavior (*Ohyama et al., 2015*). EM reconstruction identified multiple pathways between DnB and Goro, through A09e and TePn05 neurons (*Gerhard et al., 2017*)(*Figure 7A*). The DnB-A09e pathway consists of a connection between DnB and A09e neurons, which receive bilateral input from DnBs (*Figure 7A*). A09e connects with Goro via A02o 'Wave' (*Takagi et al., 2017*)and A05q neurons. The DnB-TePn05 pathway (*Figure 7A*) consists of a connection between DnBs and TePn05, which synapses with both Basin-2,4, and Basin-3 populations. Basins make extensive connections with A23g, and A05q, which synapse onto Goro (both directly and indirectly through Wave) (*Ohyama et al., 2015*). A05q links to Goro have been functionally validated (*Ohyama et al., 2015*). Thus, A09e and TePn05 networks may underlie DnB-Goro communication.

We asked whether DnBs are functionally connected to Goro rolling command-like neurons. We activated DnB neurons using Chrimson (*Klapoetke et al., 2014*), and monitored calcium responses in Goro using GCaMP6s (*Figure 7B*). We targeted Chrimson activation to the entire *412-Gal4* CNS pattern using a 630 nm LED or to 1–2 segments of *412-Gal4* neurons in the nerve cord, using a phaser module to target the multiphoton laser. Both whole CNS and segmentally targeted activation generated calcium increases in Goro axons (*Figure 7C and E*). Activating *412-Gal4* brain neurons, which do not include DnBs, did not alter Goro responses (*Figure 7D*). These results support a functional link between DnB neurons and Goro rolling command-like neurons.

Given that C-bending and rolling could be activated separately, we asked whether DnB activity might coordinate rolling through Goro command neurons, and bending through alternate pathways. To test this hypothesis, we activated DnBs while selectively silencing Goro activity (*412-Gal4$^+$* Goro$^-$) using *16E11-LexA* (*Ohyama et al., 2015*) (*Figure 7F*). As expected, control animals (*412-Gal4$^+$*) showed nociceptive behavior consisting of C-bending and rolling (61% bend→roll, 39% bend→no

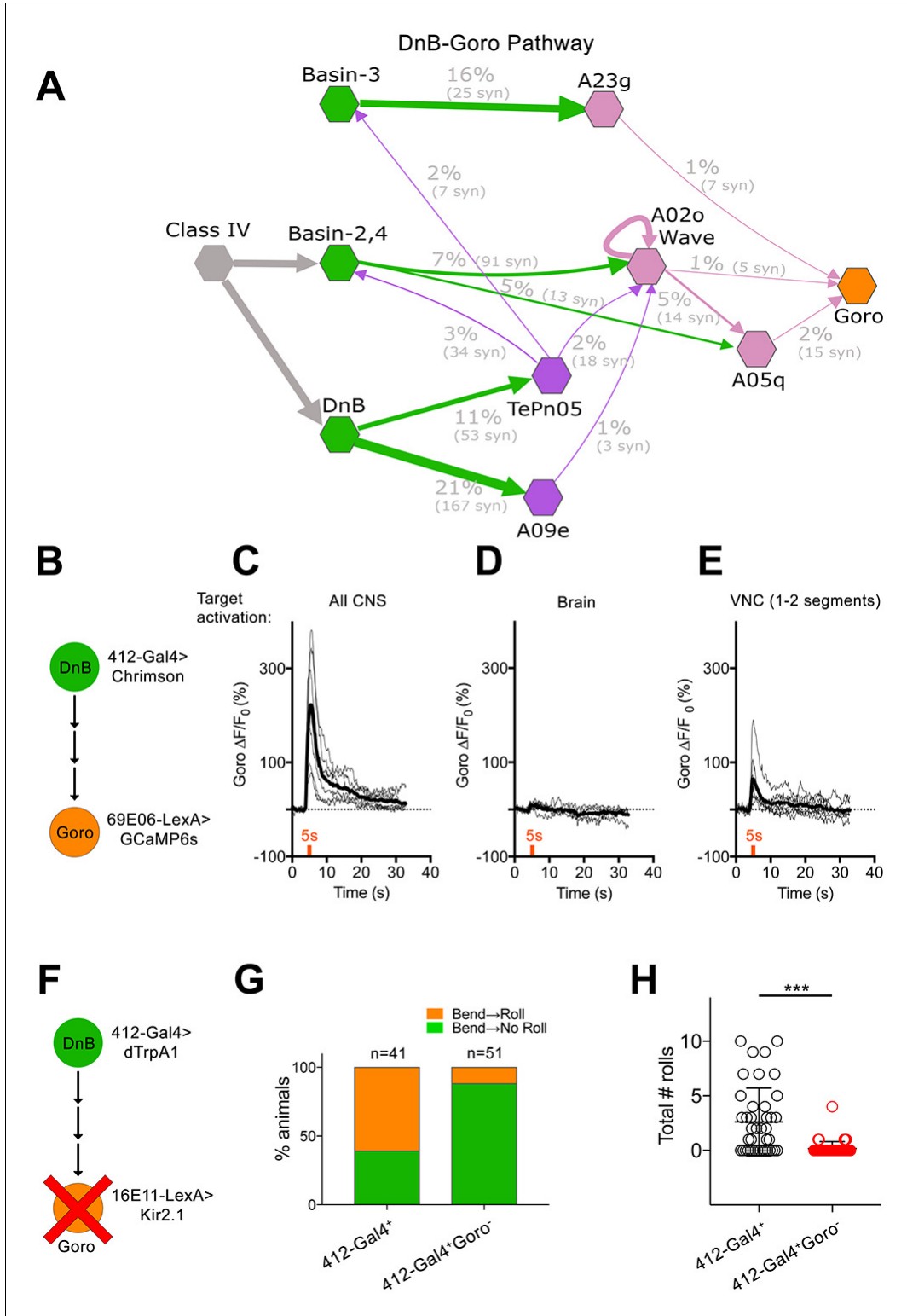

**Figure 7.** DnBs promote rolling, but not C-bending, through Goro network. (**A**) Wiring diagram of DnB to Goro rolling command-like neuron. Percentage represents fraction of total dendritic inputs provided by upstream neuron class. Percentages may underestimate contribution of neuron class due to lack of data from all segments. Number of reconstructed synapses is indicated in parentheses. Hexagons represent groups of neurons. (**B**) Experimental setup for Goro imaging experiments. Activity in DnB neurons is driven by *UAS-Chrimson* expression and optogenetic stimulation across the entire CNS (**C**), brain only (**D**) or 1–2 segments of DnB neurons in the VNC

*Figure 7 continued on next page*

*Figure 7 continued*

(**E**). GCaMP6s was targeted to Goro neurons using *69E06-LexA* to monitor calcium responses. (**C**) $\Delta F/F_0$ measured in Goro axons (n = 6) upon full CNS optogenetic activation of *412-Gal4* neurons. (**D**) $\Delta F/F_0$ measured in Goro axons (n = 4) upon brain only (lacking DnBs) optogenetic activation of *412-Gal4* neurons. (**E**) $\Delta F/F_0$ measured in Goro axons (n = 7) upon optogenetic activation of 1–2 DnB neurons. (**F**) Experimental setup for DnB thermogenetic activation and Goro silencing. (**G**) Thermogenetic activation of *412-Gal4* neurons (*412-Gal4+*) leads to dominant bend-roll nociceptive phenotype (bend→roll, orange). A minority of larvae show bending without rolling responses (bend→ no roll, green). Coincident silencing of Goro neurons (*412-Gal4+Goro-*) subdues rolling responses but does not disrupt bending. (**H**) Total number of rolls shown by control larvae (*412-Gal4* thermogenetic activation; black open circles) and upon coincident Goro silencing (red open circles). Error bars represent standard deviation. P values are indicated as *p<0.05, **p<0.01 ***p<0.001, as tested by Mann Whitney. List of neurons included at each node can be found in supplementary file. (See also ***Figure 7—figure supplement 1***).

DOI: https://doi.org/10.7554/eLife.26016.027
The following source data and figure supplements are available for figure 7:

**Source data 1.** Summary table of graph data and statistical testing for Goro functional imaging and behavior experiments.
DOI: https://doi.org/10.7554/eLife.26016.029
**Figure supplement 1.** Silencing PMSI premotor neurons reduces rolling.
DOI: https://doi.org/10.7554/eLife.26016.028
**Figure supplement 1—source data 1.** Summary table of graph data and statistical testing for PMSI silencing experiments.
Genotypes, number of animals tested, graph data and statistical testing presented for PMSI silencing experiments.
DOI: https://doi.org/10.7554/eLife.26016.030

roll)(***Figure 7G***). By contrast, *412-Gal4+* Goro- larvae showed bending behavior without rolling upon thermogenetic activation (12% bend→roll, 88% bend→no roll) (***Figure 7G***; ***Video 5***). Correspondingly, we observed a significant decrease in total number of rolls exhibited by *412-Gal4+* Goro- larvae (***Figure 7H***). These data suggest that C-bending and rolling are separable and coordinated by DnB activity to generate a rapid escape locomotion sequence.

## Discussion

Nocifensive escape behavior in *Drosophila* larvae consists of C-shaped body bending and rolling, followed by rapid forward crawling (***Hwang et al., 2007***; ***Ohyama et al., 2013***). Recent studies have begun to identify circuits that mediate nocifensive behaviors (***Kaneko et al., 2017***; ***Ohyama et al., 2015***; ***Yoshino et al., 2017***). Prior work identified Basin neurons as multisensory interneurons that drive rolling behavior in response to vibration and noxious stimuli, and identified downstream Goro as command-like neurons for rolling (***Ohyama et al., 2015***). Here, we have identified and characterized DnB interneurons that are essential for nocifensive behavior in *Drosophila* larvae (***Figure 8***). DnB neurons are direct targets of nociceptive cIV neurons and multiple mechanosensory cell types, including cII and cIII gentle touch da neurons and es neurons. Thus, DnBs provide a potential node for multisensory integration of tactile and noxious stimuli. The convergence of input from cIII gentle-touch receptors and cIV nociceptors onto DnB neurons is reminiscent of vertebrate interneurons that receive direct excitatory input from C-fiber/Aδ nociceptors and Aß mechanoreceptors (***Duan et al., 2014***). Based on these studies nociceptive inputs appear to be integrated with multiple

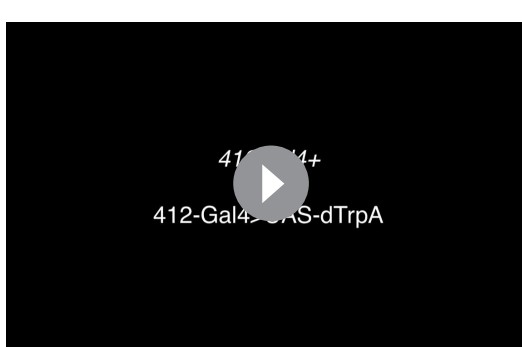

**Video 5.** Activating *412-Gal4* while silencing Goro neurons biases larvae towards bending without rolling. Video shows bending without rolling when *412-Gal4* neurons, including DnBs, are activated while suppressing Goro activity
DOI: https://doi.org/10.7554/eLife.26016.031

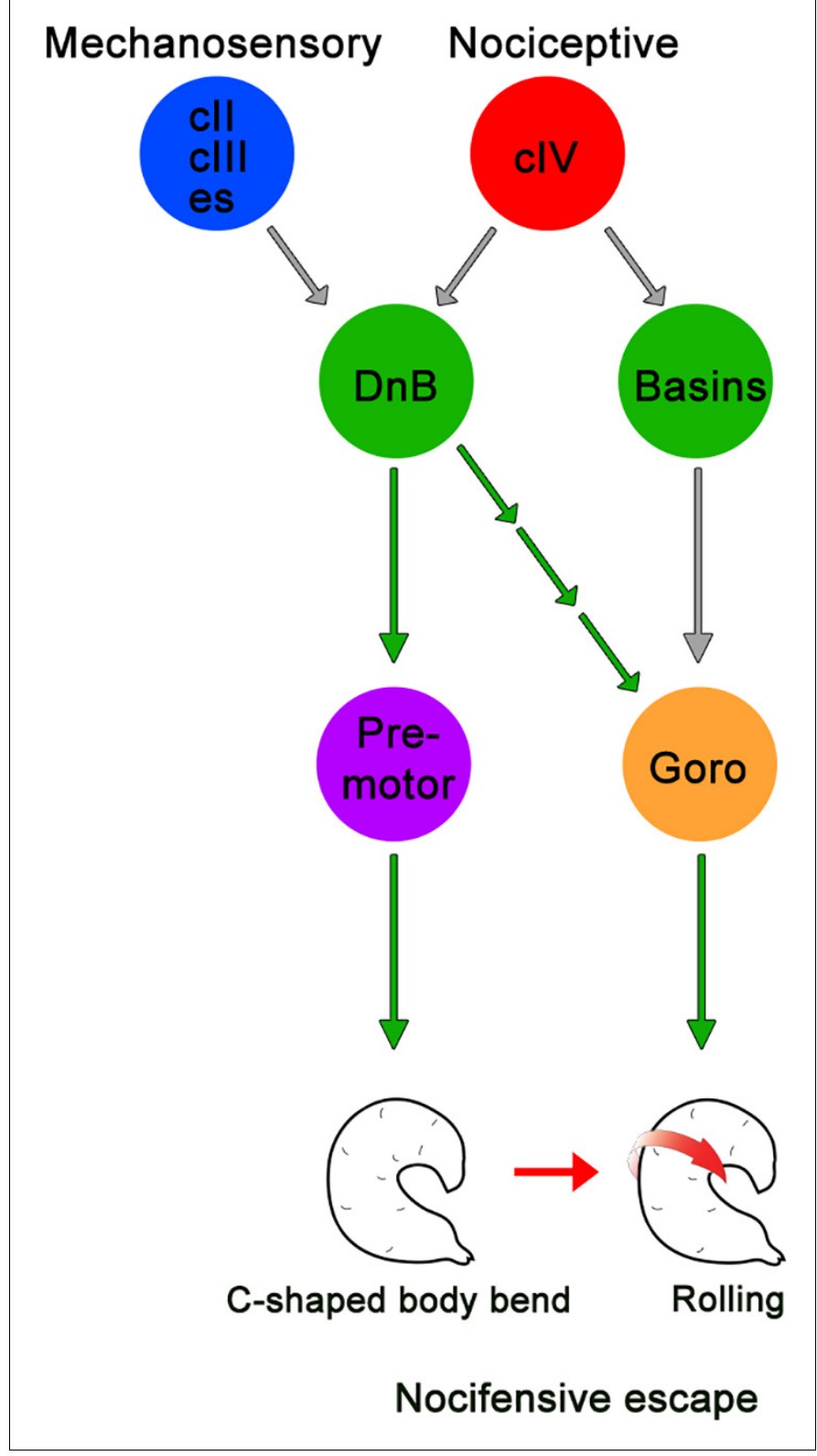

**Figure 8.** Summary model for DnB neurons controlling nocifensive escape. DnB neurons receive dual mechanosensory and nociceptive input, and promote nocifensive escape behavior via co-activation of downstream premotor circuits and command-like Goro neurons.

DOI: https://doi.org/10.7554/eLife.26016.032

mechanosensory submodalities by Basin and DnB interneurons.

EM reconstruction of DnB targets supported divergent major downstream circuitry. Output synapses on DnB axons converge on premotor neurons, at least some of which promote peristaltic wave propagation during locomotion (*Fushiki et al., 2016*; *Kohsaka et al., 2014*). Other downstream neurons receive input from presynaptic sites on the DnB dendrite, and lead to Goro rolling command-like neurons (*Ohyama et al., 2015*). The spatial segregation of DnB output sites may mirror a functional segregation of downstream circuitry into bending and rolling modules. It is still unclear which muscle groups are recruited and how segments coordinate during body bending and rolling. We provide evidence that silencing the PMSI cohort, which includes direct DnB targets A02g and A02e, reduces rolling behavior. PMSIs are glutamatergic inhibitory premotor neurons that terminate motor neuron bursting to regulate crawling speed *Kohsaka et al., 2014*). Future work to selectively silence groups of premotor neurons will help to elucidate their role in nocifensive escape downstream of DnBs. Although silencing DnB neurons slightly increased the speed of forward locomotion, overall, forward crawling remained intact. Given that peristaltic waves also consist of segmental contractions, links to premotor neurons provide candidate neurons for dual control of crawling and C-shape bending behavior. Notably, DnB neurons target motor neurons innervating LT1 muscles, which have been implicated in larval self-righting behaviors (*Picao-Osorio et al., 2015*). Self-righting consists of a C-shape type body bend, and 180° turn, so it is possible that LT1 muscles facilitate curved body bends that underlie both self-righting and rolling behavior. We note that the impact of DnB neurons on nociceptive circuits is likely to be more broad than indicated by synaptic connections, since EM and marker expression suggest that DnB neurons are peptidergic. Identification of the putative neuropeptide expressed by DnB neurons, and physiological effects, will be an important future direction, particularly given the important role of neuropeptides in vertebrate pain pathways. (*Faris et al., 1983*; *Mantyh et al., 1997*; *Ribeiro-da-Silva and De Koninck, 2008*; *Sun et al., 2004*), and recent evidence that mechanical nociception in larvae is under peptidergic control (*Hu et al., 2017*).

Prior data showed that rolling is directional and is advantageous for dislodging attacking parasitoid wasps (*Hwang et al., 2007*). Efficient rolling occurs coincident with deep C-shaped body bends, but the significance of these body bends for escape behavior has not been determined. DnB neural circuitry appears to be critically important for evoking body bend behavior prior to and during nocifensive rolling. Bending may provide the initial, most rapid, form of withdrawal from a noxious stimulus, and may subsequently support rolling locomotion by orienting and focusing the energy of muscle contraction into lateral thrusts. Re-orientation of denticle belts, triangle-shaped extensions of the cuticle, may also aid rapid lateral locomotion by providing substrate traction. Compromised escape rolling upon DnB inactivation may therefore arise both from weakened Goro activation and decreases in body bend angle. Understanding the circuit mechanisms that promote bending downstream of DnB neurons, and the muscle activities and physical mechanisms that underlie rolling behavior are important future aims.

Analysis of DnB function revealed modular control of nocifensive escape behavior, consistent with EM reconstruction data. When DnB neurons were ectopically activated we observed C-shaped body bending that was often, but not always, associated with rolling. Other, non-rolling, animals bent with minimal crawling, or bent persistently while attempting to crawl forward. These observations provided initial evidence that C-shaped bending and rolling control circuits are separable, and that nocifensive bending could be combined with other behaviors, like pausing or crawling. Our loss of function data supported bending as a primary motor output of DnB activity, with probabilistic activation of rolling motor programs. These behaviors could conceivably be linked, such that reduction in bending compromises rolling ability, or could arise from parallel influence of DnB activity on bending and rolling as suggested by EM reconstruction. Consistent with an important role for DnBs in promoting rolling, silencing Goro while activating DnB neurons promoted persistent bending without rolling, and uncoordinated snake-like forward crawling. This result further implicates a separate premotor circuitry in nocifensive body bending. These data further suggest that the bend-roll sequence must be tightly regulated by interactions between the parallel bend-roll premotor circuits, such that bending occurs first to facilitate rolling, which occurs second. However, bending can occur without being followed by rolling, indicating C-shaped bending itself is not sufficient to trigger rolling. Such independent, but sequentially regulated behavioral modules are consistent with hierarchical models of sequence generation as in fly grooming (*Seeds et al., 2014*), human speech (*Lashley, 1951*), roll-

crawl sequence (*Ohyama et al., 2013*), and hunch-bend sequence (*Jovanic et al., 2016*). We note however, that although bending and rolling are sequential, they co-occur for much of the defensive behavior sequence, in contrast to such sequential and non-overlapping behavioral sequences. Elucidating the mechanisms of timing and interaction between the different circuit modules (bend vs roll) identified therefore promises to shed light on the general mechanisms of circuit implementation of sequence generation and co-ordination between different motor modules.

# Materials and methods

## Key resources table

| Reagent type (species) or resource | Designation | Source or reference | Identifiers | Additional information |
|---|---|---|---|---|
| strain, strain background (D. melanogaster) | PB[IT.Gal4]0412 | PMID:21473015 | | |
| strain, strain background (D. melanogaster) | R70F01-LexA | PMID: 23063364 | RRID:BDSC_53628 | |
| strain, strain background (D. melanogaster) | R69E06-LexA | PMID: 23063364 | RRID:BDSC_54925 | |
| strain, strain background (D. melanogaster) | $ppk^{1.9}$-Gal4 | PMID: 12956960 | | |
| strain, strain background (D. melanogaster) | 20X-UAS-IVS-GCaMP6m | PMID: 23868258 | RRID:BDSC_42748 | |
| strain, strain background (D. melanogaster) | UAS-dTrpA1 | PMID: 18548007 | RRID:BDSC_26263; RRID:BDSC_26264 | |
| strain, strain background (D. melanogaster) | UAS-ReaChR | PMID: 23995068 | RRID:BDSC_53749; RRID:BDSC_53741 | |
| strain, strain background (D. melanogaster) | tub > Gal80>; tsh-LexA, 8X-LexAop2-FLPL/CyO-RFP-tb; UAS-10X-IVS-myr:GFP | | | Gift from Dr. Marta Zlatic |
| strain, strain background (D. melanogaster) | tub > Gal80>; tsh-LexA, 8X-LexAop2-FLPL/CyO-RFP-tb; UAS-dTrpA1/TM6B | | | Gift from Dr. Marta Zlatic |
| strain, strain background (D. melanogaster) | UAS-TNT | PMID: 7857643 | RRID:BDSC_28838 | |
| strain, strain background (D. melanogaster) | UAS-TNTi | PMID: 7857643 | RRID:BDSC_28840 | |
| strain, strain background (D. melanogaster) | tsh-Gal80 | | | Gift from Dr. Julie Simpson |
| strain, strain background (D. melanogaster) | 8X-LexAop2FLPL; 10X-UAS > Stop > myr:GFP | PMID: 24183665 | | |
| strain, strain background (D. melanogaster) | 8X-LexAop2FLPL; 10X-UAS > Stop > $Kir^{2.1}$-GFP | PMID: 24183665 | | |
| strain, strain background (D. melanogaster) | 13X-LexAop2-IVS-TNT::HA | PMID: 24507194 | | Gift from Dr. Chi-Hon Lee |
| strain, strain background (D. melanogaster) | LexAop-$Kir^{2.1}$ | PMID: 24991958 | | Gift from Dr. Barry Dickson |
| strain, strain background (D. melanogaster) | 20xUAS-CsChrimson-mCherry | PMID: 27720450 | | |
| strain, strain background (D. melanogaster) | 13xLexAop2-IVS-GCaMP6s | PMID: 23868258 | | |
| strain, strain background (D. melanogaster) | yw; Mi{PTGFSTF.0} ChATMI04508-GFSTF.0 | PMID: 26102525 | ID_BSC: 60288 | |
| antibody | anti-GFP | Abcam | RRID: AB_300798 | 1:1000 |
| antibody | anti-DsRed | Clontech | RRID:AB_10013483 | 1:250 |
| antibody | anti-Fasciclin II | DSHB | RRID:AB_528235 | 1:100 |

*Continued on next page*

Continued

| Reagent type (species) or resource | Designation | Source or reference | Identifiers | Additional information |
|---|---|---|---|---|
| antibody | anti-5HT | Sigma | RRID:AB_477522 | 1:1000 |
| antibody | anti-dvGLUT | PMID: 15548661 | RRID:AB_2314347 | 1:10,000 |
| antibody | anti-GABA | Sigma | RRID:AB_477652 | 1:100 |
| antibody | anti-ChAT | DSHB | RRID:AB_2314170 | 1:100 |

## Fly stocks

(1)*PB[IT.Gal4]0412* (referred to in the text as *412-Gal4*; (*Gohl et al., 2011*), (2) *UAS-mCD8-GFP* (*Lee and Luo, 1999*), (3) *ppk-CD4-tdTom* (*Han et al., 2011*), (4) *hsFLP;Sp/CyO;UAS > CD2>CD8* GFP (*Basler and Struhl, 1994*; *Wong et al., 2002*), (5) *UAS-BRP.short$^{mCherry}$* (*Schmid et al., 2008*) was provided by Dr. Richard Mann (Columbia University), (6) *UAS-DenMark* (*Nicolaï et al., 2010*), (7) *dTrpA1-QF* (Bloomington Stock Center), (8) *20X-UAS-IVS-GCaMP6m* (*Chen et al., 2013*), (9) *UAS-dTrpA1* (*Hamada et al., 2008*), (10) *UAS-ReaChR* (*Lin et al., 2013*). (11) *tub >Gal80>; tsh-LexA, 8X-LexAop2-FLPL/CyO-RFP-tb; UAS-10X-IVS-myr:GFP*, and (12) *tub >Gal80>; tsh-LexA, 8X-LexAop2-FLPL/CyO-RFP-tb; UAS-dTrpA1/TM6B* were a gift from Dr. Marta Zlatic (Janelia Research Campus, Virginia). (13) *UAS-TNT* and (14) *UAS-TNTi* (*Sweeney et al., 1995*), (15) *tsh-Gal80* was a gift from Julie Simpson (UCSB, California), (16) *R70F01-LexA* (*Jenett et al., 2012*), (17) *8X-LexAop2FLPL;10X-UAS > Stop > myr:GFP*, and (18) *8X-LexAop2FLPL;10X-UAS > Stop > Kir$^{2.1}$-GFP* (*Shirangi et al., 2013*)were a gift from Dr. James Truman (Janelia Research Campus, Virginia). (19) *13X-LexAop2-IVS-TNT::HA*(*Karuppudurai et al., 2014*), (20) *R38A10-LexA*(*Jenett et al., 2012*), (21) *ppk$^{1.9}$-Gal4* (*Ainsley et al., 2003*), (22) *w-; Sp/CyO; 13X-LexAop2-IVS-myr:GFP/TM3,Sb,e* (23) *Sp/CyO;nompC-LexA, 10X-LexAop2-myr-GFP/TM6B*, (24) *R16E11-LexA* (25) *R69E06-LexA*, (*Jenett et al., 2012*), (26) *LexAop-Kir$^{2.1}$* (*Feng et al., 2014*) was a gift drom Dr. Barry Dickson (Janelia Research Campus, Virginia), (27) *20xUAS-CsChrimson-mCherry* (*Jovanic et al., 2016*), (28) *PB[IT.Gal4]4051* (T. Clandinin and W. Grueber, unpublished), (29) *[IS.QF]0412*, (30) *13xLexAop2-IVS-GCaMP6s* (*Chen et al., 2013*), (31) *R38H01-Gal4* (*Jenett et al., 2012*), (32) *Trh-Gal4* (*Alekseyenko et al., 2010*) (33) *Per-Gal4* (*Kaneko and Hall, 2000*), (34) *yw; Mi{PTGFSTF.0}ChATMI04508-GFSTF.0* (*Nagarkar-Jaiswal et al., 2015*), (35) *UAS-Kir$^{2.1}$-eGFP* (*Baines et al., 2001*)

## Immunohistochemistry

Immunohistochemistry was performed essentially as described (*Matthews et al., 2007*). Third instar larvae were dissected in 1X PBS, fixed in 4% paraformaldehyde (Electron Microscopy Sciences) in 1X PBS for 15 min, rinsed three times in 1X PBS + 0.3% Triton X-100 (PBS-TX), and blocked for 1 hr at 4°C in normal donkey serum (Jackson Immunoresearch). Primary antibodies used were chicken anti-GFP (1:1000; Abcam), rabbit anti-DsRed (1:250, Clontech), mouse anti-1D4 anti-Fasciclin II (1:10; Developmental Studies Hybridoma Bank), rabbit anti-5HT (1:1000; Sigma), rabbit anti dvGLUT (1:10,000) (*Daniels et al., 2004*), rabbit anti-GABA (1:100; Sigma), mouse anti-ChAT (1:100; Developmental Studies Hybridoma Bank). Animals were incubated overnight in primary antibodies at 4°C, rinsed repeatedly in PBS-TX, and incubated overnight at 4°C in species-specific, fluorophore-conjugated secondary antibodies (Jackson ImmunoResearch) at 1:200 in PBS-TX. Tissue was mounted on poly-L-lysine coated coverslips, dehydrated in ethanol series, cleared in xylenes, and mounted in DPX (Fluka).

## Generation of clones

Single-cell FLP-out clones were generated by providing 1 hr heat shock at 38°C in late embryonic and early larval progeny from mating of stocks 1 and 4 (See Fly Stocks).

## Behavioral analysis

For behavioral analysis, flies were reared at 25 °C and tested as wandering third instar larvae. For each experiment, at least three trials, taken on separate days, were performed for each genotype. Larvae were only tested once unless otherwise noted.

## Thermogenetic activation

For *412-Gal4* dTrpA1 experiments, third instar larvae were rinsed briefly in double distilled water and placed on a 1% agarose gel heated to 31–34°C by a hot plate (Dri-bath type 17600, Barnstead Thermolyne). All other dTrpA1 experiments were performed on 1% agarose gels with 0.6% black ink (Super Black India ink, Speedball) using a peltier device (CP-031, TE technology) and temperature controller (TC-36–25-RS232, TE technology) to heat the gel to 32.5–33.5°C. Animals displaying 360° rotations were classified as 'rollers'. In *412-Gal4* VNC experiments, 'Bend-roll' was counted as coincident C-shaped bending and 360° rotation, 'bend-crawl' was counted when animals persistently bent as they crawled and did not perform straight forward crawling, and 'bend-only' behavior, was counted when animals remained in a curved posture without rolling or crawling. Trachea were used as a reference for bending and rolling categorization. Animal behavior was recorded using a Leica M50 camera along with Leica FireCam software and QuickTime screen capture for 60 s for *412-Gal4* activation, 29 s for *412-Gal4* VNC activation, and 30 s for all other activation experiments. Videos were quantified offline with experimenter blind to condition.

## Optogenetic activation

For optogenetic experiments, we tested animals in a photostimulation arena (*de Vries and Clandinin, 2013*). Flies were raised on molasses food with or without 1 mM all-*trans*-retinal (ATR). Third instar larvae were rinsed briefly in double distilled water and placed on a 100 × 15 mm petri dish containing double distilled water blended with yeast particles to facilitate nocifensive behavior (S. Mauthner, personal communication). Larvae were recorded using DALSA Falcon 4M30 four megapixel digital camera and CamStudio screen capture software with 10 s blue light off-10 second blue light on (23500 Lux). A dim red light was on for the entirety of the experiment to illuminate larvae during lights off periods (300 Lux). Animals displaying 360° rolling were counted as responders. Videos were quantified offline.

## Global activation assay

For the global activation assay, third instar *R70F01∩412* larvae were placed on a 1% agarose 0.6% black ink gel (Super Black India ink, Speedball) heated to 40°C by a peltier device (CP-031, TE technology) and temperature controller (TC-36–25-RS232, TE technology). Behaviors were recorded for 30 s using Leica M50 camera along with Leica FireCam and QuickTime screen capture. After experiments, animals were placed on microscope slide with 70% glycerol and a coverslip, and assessed for GFP expression under a fluorescence microscope. Behavior was quantified offline with experimenter blinded to genotype. Duration of the first rolling event was quantified by using the trachea as a reference to determine the completion of a 360° roll (*i.e.* frame before trachea starts to disappear as beginning of roll and frame where trachea is re-centered as completion of rolling event).

## Local heat assay

Local heat assay was performed as previously described (*Tracey et al., 2003*) with slight modifications. Soldering iron (SKU25337, Sinometer) was used as a noxious thermal probe and the temperature was set to 51.6–55.5°C by adjusting voltage using a variac (3PN1010B, Staco Energy). Digital thermometer (51 II, Fluke) with thermocouple temperature sensor was used to measure the temperature of the thermal probe. Larvae were lightly touched with thermal probe at segments 4–6 for 5 s. Animals were characterized as 'responder' if they performed 360° roll within 5 s, and 'non-responder' if they did not. Animal behavior was recorded using Leica FireCam and QuickTime screen capture. Videos were quantified offline with experimenter blind to genotype.

## Gentle touch assay

For the gentle touch assay, experiments were conducted as previously described (*Kernan et al., 1994*). Third instar larvae were rinsed off in double distilled water, then left to acclimate on 1% agar for 3 min. Animals were tested on 1% agar 100 × 15 mm petri dish and assigned a Kernan score for each behavior 0: no response, 1: hesitate, 2: anterior withdraw or turn, 3: single reverse wave, 4: multiple reverse waves. Experimenter was blind to genotypes during testing.

## Crawling speed assay

To assess crawling speed, larvae were rinsed in double distilled water and placed on a 1% agarose gel and tested for crawling speed using the Multiworm tracker (*Swierczek et al., 2011*). Larvae were tested three at a time at 25°C.

## Dose-response optogenetic experiments

For dose-response optogenetic experiments, animals were tested on the FIM (Frustrated total internal reflection based Imaging Method) table (*Risse et al., 2013*), Basler ACE four megapixel near infrared sensitivity enhanced camera equipped with CMOSIS CMV4000 CMOS sensor. Camera was equipped with LM16HC-SW lens (Kowa), and BN880-35.5 filter (Visionlighttech). IR diodes (875 nm, Conrad) were used for FTIR imaging and images were acquired using Pylon camera software (Basler). Animals were placed on 0.8% agar surface ~2 mm thick (Molecular grade, Fisher Scientific) with a ring of Green (525 nm) LED lights (WFLS-G30 × 3 WHT, SuperBright LEDs) around five inches in diameter placed directly underneath the FIM table, with a standard barrel connector and a pulse-width modulation circuit based LED dimmer (CPS-F2ST; LDK-8A, SuperBrightLEDs) for light intensity control. Larvae were raised on molasses food with 1 mM all-transretinal (ATR). Third instar larvae were rinsed briefly in double distilled water and tested ~3–5 animals at a time. Animals were recorded for at least 1 s before light stimulation, and then for at least 10 s following lights ON. Trials were recorded at different light intensities: Lowest (~45 lx), Low (~200 lx), Moderate (~850 lx) and Highest (~1450 lx). Videos were collected at 10fps and quantified offline with experimenter blind to the manipulation. Only 11 s of behavior were scored per trial (1 s pre-stimulus, 10 s lights ON). Behaviors quantified: Crawling, segmental waves visible; Pausing, no movement straightened body; Bending, animal curved or on its side, and Rolling, 360° turns using bright trachea as a reference. Bending angles were quantified using the FIMTracker software (*Risse et al., 2013*)

## Calcium imaging

### DnB neurons

Calcium imaging was performed in a partially dissected larval preparation. Wandering third instar larvae were immersed in ice-cold hemolymph-like saline 3.1 (HL3.1) (70 mM NaCl, 5 mM KCl, 1.5 mM $CaCl_2$, 4 mM $MgCl_2$, 10 mM $NaHCO_3$, 5 mM Trehalose, 115 mM Sucrose, and 5 mM HEPES, pH 7.2) (*Feng et al., 2004*). The body wall of the larva was cut at segment A2 or A3 to expose the central nervous system, leaving the posterior larval body and ventral nerves intact. Dissected larvae were then transferred to an imaging chamber filled with HL3.1 equilibrated to room temperature (23–25°C). The CNS was covered with a strip of parafilm and gently pressed down onto a coverslip for immobilization during imaging. DnB neurons in the ventral nerve cord were imaged using a Zeiss LSM5 Live confocal microscope with a 20x/0.8 Plan-Apochromat objective equipped with a piezo focus drive (Physik Intrumente). Three-dimensional time-lapse imaging was performed with X-Y dimensions of 256 × 256 pixels, a slice thickness of 7 μm, 8–11 Z slices (covering 49 to 63 μm), a scan speed of 31 μsec per pixel, and 8 bit depth. The acquisition rate of Z stack images with this setting was 4 to 5 Hz. During imaging, a thermal ramp was applied locally to hemisegments A5 to A7 of the dissected larvae using a custom-made thermal probe. The temperature of the thermal probe was controlled by changing the voltage through a variac transformer (RSA-5E, Tokyo Rikosha). 15V was used to heat the probe and no voltage was applied during cooling. A t-type thermocouple probe wire (0.2 mm dia., Sansho) was placed inside of the thermal probe to monitor the temperature of the probe. Temperature data measured by the thermocouple probe were acquired at 4 Hz through a USB-TC01 digitizer (National Instruments) and recorded using the NI Signal Express software (National Instruments). The acquired images and temperature data were analyzed using MAT-LAB (Mathworks). The average of the lowest 10% fluorescent intensity was used as baseline F ($F_0$) for each region of interest, and percent fluorescent change from the baseline ($\Delta F/F_0$) was calculated for each time point. Regions of interest (ROIs) were selected as circular areas with a diameter of 6 pixels that contain the cell bodies of the DnB neurons in the maximum intensity projections of the time-series images. Probe temperature for each image frame was estimated by a linear interpolation from the raw probe temperature trace, due to differences in sampling rate and timing across images and probe temperature.

### Goro neurons

For activation of presynaptic neurons (Down and Back) with CsChrimson and imaging in Goro neurons, the central nervous system of wandering third instar larvae was dissected in cold physiological saline, Baines solution (*Baines et al., 2001*) containing (in mM) 103 NaCl, 5 KCl, 5 HEPES, 26 NaHCO$_3$, 1 NaH2PO$_4$, 5 Trehalose, 6 Sucrose, 2 CaCl$_2$ 2H$_2$O, 8 MgCl$_2$ 6H$_2$O, and kept stable by sticking them on poly-L-lysine (SIGMA, P1524) coated cover glass placed in small Sylgard (Dow Corning) plates. 620 nm LED (Mightex Systems Inc.) was used for whole CNS CsChrimson activation and 1040 nm laser using Phaser module (Intelligent Imaging Innovations, Inc.) for localized CsChrimson activation. We imaged the axon of Goro neurons. Image data were processed by ImageJ software (NIH) and analyzed using custom code written in MATLAB (Mathworks). Specifically, regions of interest (ROIs) were determined by averaging the 10 frames before stimulation and segmenting these data by the function MEAN83 in ImageJ. The mean intensity of ROI was measured in ImageJ. In all cases, changes in fluorescence were calculated relative to baseline fluorescence levels (F$_0$) as determined by averaging over a period of at least 3 s just before CsChrimson activation. $\Delta F/F_0$ values were calculated as $\Delta F/F_0 = (Ft - F_0) / F_0$, where Ft is calculated by subtracting the background fluorescence from the fluorescence mean value of a ROI in a given frame.

### Boundary curvature and kymograph analysis

Larval curvature was determined as previously described (*Driscoll et al., 2011*; *Driscoll et al., 2012*) with modifications. Frames were extracted from 30 fps videos and thresholded. A size filter was applied to remove artifacts and debris. Artifacts closely associated with the animal (such as light specks or motion blurs) that would interfere with extraction of boundary curvature were manually removed blind to treatment by painting over the artifact with the background color (black). The boundary shape of the animal was parametrized with 300 boundary points. At each boundary point, we calculated the curvature by fitting a circle to that point and two points that are 10 boundary points away from it. Curvature Index (C.I.) was defined as the reciprocal of the radius of that circle so that smaller circles (greater curvature) had a higher absolute C.I. value. If the midpoint of the line segment joining these two flanking points is outside the larval outline, the C.I. was assigned to be positive (concave curvature); otherwise it was assigned to be negative (convex curvature). For visualization, a color scale was generated with warm colors corresponding to positive C.I. (i.e. concave curvature segment) and cool colors corresponding to negative C.I. (i.e. convex curvature segment). Kymographs were generated by plotting curvature index (colored by magnitude) of 300 boundary points across time. Alignment of the 300 points across time in kymographs was achieved by mapping points across frames to minimize the sum of the square distance of points between successive frames. To maintain the relative head and tail positions in the kymographs, we manually corrected for misalignment. Animals from *R70F01∩412*-silenced and non-silenced groups were selected for boundary curvature analysis if they fulfilled one of two criteria (1) completed rolling (360° turns), or (2) 'attempted rolling' (i.e. exhibited lateral body turns that were <360°; trachea was used as a reference to assess lateral turning). Classification was performed blind to genotype. An identical number of animals were analyzed for each treatment, which for the non-silenced animals corresponded to the first 24 animals tested. Custom MATLAB scripts were used for curvature analyses and generation of kymographs.

### Quantification of boundary curvature

Quantification was focused on boundary points with positive C.I. values, which reflects concave curvature (i.e. mainly inside C-shaped bend). To further refine analyses, curvature indices (C.I.) taken at boundary points along the body were included, with the exception of the head and tail (defined to be within 25 points of the head and tail tip points) as their curvature reflected the animal's shape at the tips, and not the curvature of the animals' body. Percent of boundary points at low curvature (0 < C.I.<0.027) and high curvature (C.I. > 0.027) were compared between control and *R70F01∩412*-silenced animals. The C.I. cutoff for low curvature vs. high curvature was defined as the median of the C.I. in the control group. Multivariate analysis of variance (MANOVA) was performed with Bonferroni correction, for multiple testing, followed by post-hoc T-test to determine exact p-value.

## EM reconstruction of DnB circuits

EM reconstruction was performed using CATMAID as previously described (*Ohyama et al., 2015*; *Schneider-Mizell et al., 2016*). A09l (DnB) neurons in A1 were identified during circuit reconstruction downstream class IV sensory neurons (*Ohyama et al., 2015*), and were verified as *412-Gal4* labeled neurons based on morphology and cell body position. DnB annotated synapses then served as starting points to reconstruct the pre- and post-synaptic connectome.

## Statistical analysis

For categorical data analysis (i.e. responder vs. nonresponder), we utilized Fisher exact test or Chi square test (if expected value=<5) followed by Bonferroni correction if multiple testing was used. When comparing two groups of quantitative data (e.g. number of rolls), unpaired t-test was performed if data showed a normal distribution (determined using D'Agostino and Pearson omnibus normality test) and Mann-Whitney test if data distribution was non-normal.When comparing three or more groups, data were analyzed using One-way ANOVA or Kruskal-Wallis test with Dunn's correction for multiple testing, followed by post-hoc T-test to determine exact p-value.

## Acknowledgements

We are grateful to Drs. Richard Axel, Tom Clandinin, Barry Dickson, Toshiro Kitamoto, Chi-Hon Lee, Troy Shirangi, Julie Simpson, and James Truman for fly stocks. We thank Thomas Kahn for writing a custom shell script for blinding video files. We thank Rick D Fetter and the Fly EM Project Team at HHMI Janelia for the gift of the EM volume, and HHMI Janelia for funding. We thank Hiroshi Kohsaka, Akinao Nose, Ingrid Andrade, Javier Valdes-Aleman, Akira Fushiki, Aref Arzan Zarin, Maartin Zwart and Casey Schneider-Mizell for neuron reconstructions. We additionally thank Javier Valdes-Aleman for neuron reconstruction training. We are grateful to the Clandinin and Grueber labs for contributing to screening and annotating of the InSITE Gal4 lines. We thank Drs. Peter Soba and Rebecca Yang for communication of unpublished results. We thank Tanya Tabachnik (ZI Advanced Instrumentation) and Darcy Peterka (ZI Cellular Imaging) for their advice in FTIR imaging. We thank Drs. Emiko Suzuki and Akatsuki Kimura for assisting the use of LSM5 Live microscope in the calcium imaging experiment. The 1D4 anti-Fasciclin II antibody was developed by Dr. Corey Goodman and the ChAT4B1 antibody was developed by Dr. Paul Salvaterra and obtained from the Developmental Studies Hybridoma Bank, created by the NICHD of the NIH and maintained at The University of Iowa, Department of Biology, Iowa City, IA 52242. Research reported in this publication was supported by the Thompson Family Foundation Initiative in CIPN and Sensory Neuroscience at Columbia University Medical Center (WBG), a National Science Foundation Graduate Research Fellowship (AB), National Institutes of Health (NIH) Predoctoral Fellowship 1F31NS090909-01 (AB), Columbia University, NIH R01 NS061908 (WBG), NIH R24 NS086564 (T Clandinin and WBG), NIH R01 GM086458 (WDT), National Institute of Genetics Postdoctoral Fellowship (KH) and JSPS KAKENHI 26890025 (KH).

## Additional information

### Funding

| Funder | Grant reference number | Author |
|---|---|---|
| National Science Foundation | Graduate Research Fellowship | Anita Burgos |
| National Institutes of Health | NS090909-01 | Anita Burgos |
| Japan Society for the Promotion of Science | KAKENHI 26890025 | Ken Honjo |
| Thompson Family Foundation | Innovation Award | Grace Ji-eun Shin Wesley B Grueber |
| National Institutes of Health | GM086458 | W Daniel Tracey |
| Howard Hughes Medical Institute | | Marta Zlatic Albert Cardona |

| National Institutes of Health | NS061908 | Wesley B Grueber |
| National Institutes of Health | NS086564 | Wesley B Grueber |

The funders had no role in study design, data collection and interpretation, or the decision to submit the work for publication.

## Author contributions

Anita Burgos, Conceptualization, Formal analysis, Funding acquisition, Validation, Investigation, Visualization, Methodology, Writing—original draft, Project administration, Writing—review and editing; Ken Honjo, Tomoko Ohyama, Formal analysis, Investigation, Methodology, Writing—review and editing; Cheng Sam Qian, Software, Formal analysis, Methodology, Writing—review and editing; Grace Ji-eun Shin, Resources, Methodology; Daryl M Gohl, Marion Silies, Resources, Writing—review and editing; W Daniel Tracey, Supervision, Funding acquisition, Writing—review and editing; Marta Zlatic, Resources, Supervision, Funding acquisition, Writing—review and editing; Albert Cardona, Resources, Data curation, Supervision, Funding acquisition, Validation, Methodology, Writing—review and editing; Wesley B Grueber, Conceptualization, Supervision, Funding acquisition, Visualization, Writing—original draft, Project administration, Writing—review and editing

## Author ORCIDs

Anita Burgos https://orcid.org/0000-0003-4603-2086
Cheng Sam Qian http://orcid.org/0000-0002-2456-3153
W Daniel Tracey http://orcid.org/0000-0003-4666-8199
Albert Cardona http://orcid.org/0000-0003-4941-6536
Wesley B Grueber http://orcid.org/0000-0001-6751-256X

## Decision letter and Author response

Decision letter https://doi.org/10.7554/eLife.26016.037
Author response https://doi.org/10.7554/eLife.26016.038

## Additional files

### Supplementary files

• Source code 1. Larval body curvature analysis.
DOI: https://doi.org/10.7554/eLife.26016.033

• Source code 2. Generate kymograph from curvature analysis.
DOI: https://doi.org/10.7554/eLife.26016.034

• Supplementary file 1. Main Figure genotypes.
DOI: https://doi.org/10.7554/eLife.26016.035

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
