## [Decision Letter]

Thank you for submitting your article "Nociceptive interneurons control modular motor pathways to promote escape behavior in *Drosophila*" for consideration by *eLife*. Your article has been favorably evaluated by K VijayRaghavan (Senior Editor) and three reviewers, one of whom is a member of our Board of Reviewing Editors. The reviewers have opted to remain anonymous.

The reviewers have discussed the reviews with one another and the Reviewing Editor has drafted this decision to help you prepare a revised submission.

Summary:

This paper asks how noxious stimuli are transformed into escape behaviors by performing a circuit-level analysis in *Drosophila* larvae. Strong mechanical and high thermal stimulation induce C-shaped body bending and rolling behavior, followed by rapid forward locomotion. We know the class IV DA neurons are required for these responses, but the circuits downstream of this nociceptor have not been fully worked out. Recent studies have examined elements of this downstream circuit (Ohyama et al., 2015 and Jovanic et al., 2016) but those studies did not examine how the bending, rolling, and forward locomotion are generated in sequence. The authors identify a new neuron (Dnb) required for the escape response to noxious stimuli, and then use a comprehensive set of state-of-the-art genetic tools (activation, silencing, functional connectivity, etc.) combined with EM circuit reconstruction to determine which neurons are upstream and downstream of Dnb and how this circuit organization predicts the behavioral sequence produced in response to noxious stimuli. Overall this is a comprehensive, well-written story with a lot of interesting details. While the reviewers are supportive of publication, there are concerns that should be addressed and some suggestions for improvement.

Major concerns and suggestions:

1) We note that neurons that function downstream of the DnB neuron were incompletely analyzed. Specifically, there are many weakly connected neurons that are postsynaptic to DnB, but little functional analysis was done to address their functions. The only functional analysis done was to monitor or silence Goro neurons, which are not a direct downstream target of DnB and has previously been shown to be required for rolling. All of the additional neurons downstream of DnB, having 3 or more synapses, and not tested in the context of the rolling/bending behavior are: A27k, A01x1, A18l, A02g, A02e, A03g, A10a, A01c, A09a, A10f, TePn05. While not an absolute requirement for publication, the impact of this manuscript would be greatly enhanced with additional analysis of one or more of these downstream neurons; Additional experiments that address this issue would be welcome.

2) In Figure 2 the authors show that *412-Gal4* VNC activation can induce body bending both with and without rolling, whereas cIVDA activation caused bending and rolling together. To be more quantitative, it would help to have dose response curves for the amount of time spent doing each of the 3 behaviors (B, B+360, B+C) relative to the intensity of optogenetic activation. We understand these experiments were done with trpA1 initially, but the differences in phenotypes could either say something about the role of each of these neurons in the behavior or it could simply be due to levels of activation. In addition, the authors could examine the intensity of the behaviors (not just amount of time spent doing the behavior, but strength of each behavior…as they start to get at in Figure 3 when they examine the amount of C-shaped bending). Also, to directly confirm that C-IV neurons excite DnB neurons, the authors could activate 38A10 (C-IV neurons) optogenetically or using TRP and monitor GCaMP in DnB neurons.

3) The GRASP experiment does not control for neuron-neuron contact versus pre-post synapse contact. Without ruling out a false positive signal from neuron-neuron contact this experiment is difficult to interpret. We don't think the result is most important, given the other findings, and it may be best to delete this result rather than spending more time doing the appropriate controls.

---

## [Author Response]

Major concerns and suggestions:1) We note that neurons that function downstream of the DnB neuron were incompletely analyzed. Specifically, there are many weakly connected neurons that are postsynaptic to DnB, but little functional analysis was done to address their functions. The only functional analysis done was to monitor or silence Goro neurons, which are not a direct downstream target of DnB and has previously been shown to be required for rolling. All of the additional neurons downstream of DnB, having 3 or more synapses, and not tested in the context of the rolling/bending behavior are: A27k, A01x1, A18l, A02g, A02e, A03g, A10a, A01c, A09a, A10f, TePn05. While not an absolute requirement for publication, the impact of this manuscript would be greatly enhanced with additional analysis of one or more of these downstream neurons; Additional experiments that address this issue would be welcome.

We have analyzed the potential role of A02e and A02g neurons, part of the *period*-positive median segmental interneuron cohort (PMSIs) (Kohsaka et al., 2014; Kohsaka et al., 2017). Upon silencing PMSIs, we found that silenced animals showed reduced rolling (Figure 7— figure supplement 1), suggesting lower rolling efficiency. More selective manipulations will be necessary to explore the basis for the rolling deficit and to assign these deficits to specific neurons within the PMSI population.

2) In Figure 2 the authors show that 412-Gal4 VNC activation can induce body bending both with and without rolling, whereas cIVDA activation caused bending and rolling together. To be more quantitative, it would help to have dose response curves for the amount of time spent doing each of the 3 behaviors (B, B+360, B+C) relative to the intensity of optogenetic activation. We understand these experiments were done with trpA1 initially, but the differences in phenotypes could either say something about the role of each of these neurons in the behavior or it could simply be due to levels of activation. In addition, the authors could examine the intensity of the behaviors (not just amount of time spent doing the behavior, but strength of each behavior…as they start to get at in Figure 3 when they examine the amount of C-shaped bending). Also, to directly confirm that C-IV neurons excite DnB neurons, the authors could activate 38A10 (C-IV neurons) optogenetically or using TRP and monitor GCaMP in DnB neurons.

We performed the suggested dose response experiment to study the roles of the DnB and class IV da neurons in nociceptive behavior. We optogenetically activated these populations at four different light intensities and monitored bending, rolling, crawling and pausing during 10 seconds of activation. We found that while cIV optogenetic activation triggered rolling across activation intensities, *412-Gal4* activation of DnB neurons triggered bending vs. bending→rolling modules in a dose-dependent manner (Figure 2).

Relevant to the second point, we measured the degree of bending across levels of activation and found that cIV activation triggered rapid changes in bending which coincided with periods of rolling (Figure 2B), whereas *412-Gal4* activation results in a ramping of bending intensity, which overlaps with rolling events during higher levels of activation (Figure 2D).

Relevant to whether cIV neurons directly excite DnB neurons, in our original submission we aimed to determine whether DnB neurons respond to noxious stimulation in a cIV dependent manner. We found that reducing cIV activity decreased calcium responses in DnB somas during noxious stimulation compared to control. Our interpretation of these results is that DnB responses to noxious stimulation are dependent on cIV nociceptor activity (Figure 3A-F), which given our EM data is likely due to direct inputs.

3) The GRASP experiment does not control for neuron-neuron contact versus pre-post synapse contact. Without ruling out a false positive signal from neuron-neuron contact this experiment is difficult to interpret. We don't think the result is most important, given the other findings, and it may be best to delete this result rather than spending more time doing the appropriate controls.

We have taken the advice of the reviewers and removed our GRASP data given the redundancy with EM data.